# Geometry of Long-Tailed Representation Learning: Rebalancing Features for Skewed Distributions

**Lingjie Yi**[1],[*] **Jiachen Yao**[1], **Weimin Lyu**[1], **Haibin Ling**[1], **Raphael Douady**[2], **Chao Chen**[1]
[1] Stony Brook University, [2] University Paris 1 Pantheon-Sorbonne

## Abstract

Deep learning has achieved significant success by training on balanced datasets. However, real-world data often exhibit long-tailed distributions. Empirical studies have revealed that long-tailed data skew data representations, where head classes dominate the feature space. Many methods have been proposed to empirically rectify the skewed representations. However, a clear understanding of the underlying cause and extent of this skew remains lacking. In this study, we provide a comprehensive theoretical analysis to elucidate how long-tailed data affect feature distributions, deriving the conditions under which centers of tail classes shrink together or even collapse into a single point. This results in overlapping feature distributions of tail classes, making features in the overlapping regions inseparable. Moreover, we demonstrate that merely empirically correcting the skewed representations of the training data is insufficient to separate the overlapping features due to distribution shifts between the training and real data. To address these challenges, we propose a novel long-tailed representation learning method, FeatRecon. It reconstructs the feature space to arrange features from different classes into symmetrical and linearly separable regions. This, in turn, enhances the model's robustness to long-tailed data. We validate the effectiveness of our method through extensive experiments on the CIFAR-10-LT, CIFAR-100-LT, ImageNet-LT, and iNaturalist 2018 datasets.

## 1 Introduction

Deep learning models have achieved significant success by training on large-scale, artificially balanced datasets (i.e., ImageNet (Deng et al., 2009)). However, in real-world scenarios, datasets often exhibit long-tailed distributions, characterized by highly imbalanced *class distribution* (i.e., the sample sizes of different classes). A few classes (called *head classes*) have a large number of samples, whereas many other classes (called *tail classes*) contain only a few samples. Training on such datasets distorts a model's feature representations and decision boundaries, limiting the model's generalization capability and performance on test data.

Our understanding of balanced data representation has advanced significantly. For example, using the powerful representation learning tool, *contrastive learning* (Khosla et al., 2020), it has been shown (Graf et al., 2021) that for balanced data, when the *supervised contrastive loss* (SC loss) reaches its minimum, the feature representations of each class converge at their respective class centers, and all class centers collectively form a regular simplex (see Theorem 1 and Fig. 1a). This highly symmetrical configuration ensures separation among different classes, resulting in strong classification performance.

However, the optimal representation configuration for imbalanced data remains poorly understood. Empirical studies have revealed that, when data follow a long-tailed distribution, the optimal representations form an asymmetrical configuration, with head classes dominating the feature space. Although several methods (Zhu et al., 2022; Kang et al., 2021; Li et al., 2022) have attempted to correct this asymmetry, they primarily rely on empirical adjustments. Crucially, none of these methods

---
*Email: chris.yi@stonybrook.edu

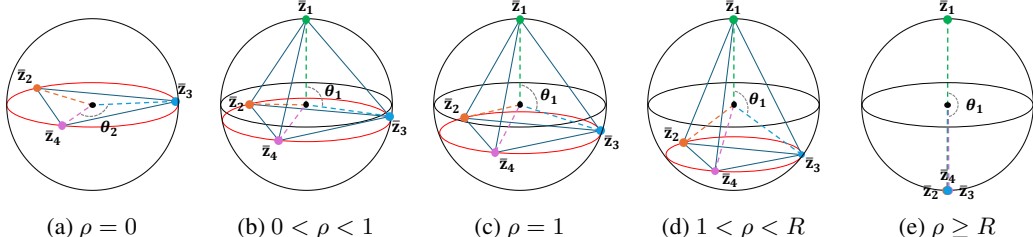

$$(a)\ \rho = 0 \qquad (b)\ 0 < \rho < 1 \qquad (c)\ \rho = 1 \qquad (d)\ 1 < \rho < R \qquad (e)\ \rho \geq R$$

Figure 1: Geometry of the optimal representation configuration for four classes with different imbalance factors, $\rho$. Centers of the four classes ($\bar{z}_1, \ldots, \bar{z}_4$) are positioned on a unit hypersphere. Assume classes 2, 3 and 4 have the same size, $N_2 = N_3 = N_4$. Class 1's size is their size multiplied by $\rho$, $N_1 = \rho N_2$. **(a)**: when class 1 is empty ($N_1 = 0$), classes 2, 3 and 4 form a regular simplex. **(b) to (e)**: As $\rho$ increases and $N_1$ increases, $\bar{z}_2$, $\bar{z}_3$ and $\bar{z}_4$ are pushed away from the equator and eventually collapse. **(c)**: when $\rho = 1$, all four classes form a regular simplex. $R$ is the critical constant at which the collapse happens (see Sec. 3 for details). $\theta_1$ is the angle between the head class center and tail class centers, and $\theta_2$ is the angle between tail classes.

provides a theoretical explanation of why and to what extent head classes dominate the feature space. Understanding this phenomenon could offer deeper insights into learning better representations of long-tailed data and inspire novel methods.

In this paper, we study the long-tailed data representation. We begin by establishing the first theoretical framework (in Theorem 2) to analyze the geometry of the optimal representation configuration, i.e., the arrangement of class centers when the SC loss is minimized, under different class distributions. In particular, we derive the analytical relationship between the imbalance factor, i.e., the ratio of sample sizes between head and tail classes, and the angles between different class centers at the optimal configuration. We show that as the imbalance factor increases, the head class increasingly dominates the feature space, pushing centers of the tail classes closer together. Beyond a certain critical threshold, centers of the tail classes collapse into a single point. Fig. 1 illustrates the optimal configuration of four classes. From Fig. 1a to Fig. 1e, as the imbalance factor continuously increases, the center of the head class ($\bar{z}_1$) pushes the other three tail classes' centers ($\bar{z}_2$, $\bar{z}_3$ and $\bar{z}_4$) closer and eventually collapse (Fig. 1e).[1]

Our theory provides insights into how long-tailed data hurts the representation learning. Without any mitigation strategy, tail classes are pushed close to one another or even collapse, resulting in overlapping feature distributions and poor separability between them. To address this issue, existing methods often readjust the empirical tail class centers to a symmetric configuration. However, due to the limited sample sizes of tail classes, these approaches may overcorrect the issue, forcing the true centers of the tail classes to be too close to the head class, leading to overlapping distributions and poor separability between head and tail classes.

In this paper, we introduce **FeatRecon**, a novel method to learn long-tailed representations. To rebalance the skewed feature distributions, this method reconstructs the feature space so that features of different classes are arranged into symmetrical and linearly separable regions. Inspired by theoretical analysis, our method addresses the center-skew issue by rebalancing the sample sizes across all classes. This is accomplished by generating synthetic features for tail classes and using them for representation learning. To ensure linear separability, the synthetic features of each class are restricted within an estimated *confidence support*, i.e., the feature space region covering most samples in that class. We derive the necessary conditions for the confidence supports to ensure that they do not overlap at the optimal configuration.

Estimation of confidence support is crucial to our method. Direct estimation of the feature distributions is challenging due to the non-Euclidean geometry of the normalized feature space and the limited sample size of tail classes. Instead, we estimate the confidence support using the center of each class and a single "central angle" parameter. Since the estimation for tail classes can be unreliable, the statistics of tail classes are regularized using those of head classes. By iteratively generating synthetic

---

[1]For completeness, our analysis encompasses the cases when class 1's size, $N_1$, is smaller than the others'. Technically, class 1 is no longer the head class when $N_1 \leq N_2 = N_3 = N_3$.

features to fill these confidence supports, adjusting representations, and re-estimating confidence supports, we can learn a feature space where both head and tail classes are equally separated, with no overlap between their confidence supports.

Our contributions are summarized as follows:

- We study the geometry of long-tailed data representation and explain how long-tailed data skew representations and limit a model's generalization capability.

- We develop a novel algorithm to generate synthetic features to balance sample sizes among all classes. The synthetic features are constrained within confidence supports which are estimated with head classes regularization.

- We propose an iterative approach to learn a symmetric and linearly separable feature space for long-tailed data. Our method iteratively generates synthetic features, adjusting representations, and re-estimates confidence support.

- We validate our method with experiments on commonly used datasets. FeatRecon outperforms SOTA performance compared to widely adopted long-tailed learning baselines.

## 2 RELATED WORK

### 2.1 LONG-TAILED RECOGNITION

*Resampling* (Byrd & Lipton, 2019) and *re-weighting* (Cui et al., 2019; Jamal et al., 2020; Chen et al., 2023) are two classical methods in long-tailed learning. The former method balances the number of training samples across different classes by either oversampling the tail classes or downsampling the head classes. The latter method balances the per class contributions to the loss function by assigning higher weights to classes with smaller gradients. Other methods emphasize adjusting decision boundaries through either *post-hoc weight normalization* (Dang et al., 2024) or *margin adjustment* (Cao et al., 2019; Menon et al., 2021; Khan et al., 2019). The former type adjusts the decision boundaries by modifying the weight norms of the matrix of classifiers, while the latter type increases the margins of the tail classes.

Recent works also explore ideas in *data augmentation* (Ahn et al., 2023; Gao et al., 2024), which adjusts the strength of class-wise augmentation to help learn class-balanced representations. *Transfer learning* (Chen & Su, 2023; Zhang et al., 2023) improve tail class learning by leveraging information from the nearby head classes. A common way for transfer learning is to assume that data follow a multivariate Gaussian distribution and transfer distribution statistics. However, robust estimation of parameters (i.e., the $K \times K$ covariance matrix) can be challenging given the small sample sizes of tail classes, and the distributional assumption does not hold for normalized features that contrastive learning deals with.

### 2.2 CONTRASTIVE LEARNING FOR LONG-TAILED DATA

*Contrastive learning* (He et al., 2020; Chen et al., 2020; Caron et al., 2020; Chen & He, 2021; Grill et al., 2020; Wang & Isola, 2020) has made great progress as a tool for representation learning. Supervised contrastive learning (SCL) (Khosla et al., 2020), by optimizing supervised contrastive loss, learns a symmetrical feature space in which the representations of each class collapse to the vertices of a regular simplex (Graf et al., 2021).

Recent studies in *long-tailed learning* (LTL) (Wang et al., 2021; Cui et al., 2021; 2023; Xuan & Zhang, 2024) incorporate an SCL module into the LTL framework, aiming to learn better representations and train most robust classifiers. However, directly using SCL is not ideal, as some (Li et al., 2022; Zhu et al., 2022) have demonstrated that SCL skews the feature space when training on long-tailed data. Many methods have since focused on empirically readjusting these skewed representations. TCL (Li et al., 2022) addresses this by predefining well-separated empirical centers. Other methods rebalance the number of contrastive pairs in the SC loss, i.e, positive contrastive pairs (Kang et al., 2021), negative contrastive pairs (Zhu et al., 2022). Our method balances contrastive pairs by generating new features for the tail classes on the surface of a unit hypersphere which are constrained within different separable hyperspherical caps.

# 3 THEORETICAL ANALYSIS: LONG-TAILED DATA SKEW CONTRASTIVE FEATURE REPRESENTATIONS

In this section, we study how long-tailed data skew the feature space. To understand how varying class distributions influence representations, we provide a theoretical framework (in Sec. 3.2) to investigate the optimal representation configuration when SC loss is minimized (see Fig. 1).

We show (in Theorem 1), for balanced data, the optimal representations form a regular simplex. This reveals that representations of different classes are equally separated to the largest extent. However, for imbalanced data, the optimal representation configuration becomes far more complex. Therefore, we focus on the simplified one-vs-all scenario. We adjust the sample size of class 1, while assuming that the remaining $K - 1$ classes have equal and fixed sample size. In Theorem 2, we study the dynamic of the geometry of the optimal representation configuration as the imbalance factor changes.

## 3.1 PRELIMINARIES

Suppose that we have $N$ training samples, $X = (x_1, \ldots, x_N) \in (\mathcal{X})^N$, randomly drawn from $K$ distinct classes, with labels $Y = (y_1, \ldots, y_N) \in (\mathcal{Y})^N$ and $\mathcal{Y} = [K] = \{1, \ldots, K\}$. A unit hypersphere (in $\mathbb{R}^h$) is defined as $\mathbb{S}^{h-1} = \{z \in \mathbb{R}^h : \|z\| = 1\}$. An encoder is a map $\varphi : \mathcal{X} \to \mathbb{R}^h$ that extracts representations from data, denoted $Z = (\varphi(x_1), \ldots \varphi(x_N))$.

In practice, contrastive learning is conducted batch-wise due to memory limitations. To simplify our analysis, we assume unlimited memory to train on all samples in a single batch. We denote the set of indices of all samples as $B = [N] = \{1, \ldots, N\}$, and the set of indices of samples of the class $k$ as $B_k = \{i : i \in B, y_i = k\}$. Let $N_k$ be the number of samples from class $k$, $N_k = |B_k|$ and $N = \sum_{k=1}^{K} N_k$. The following definitions are necessary for the study.

**Definition 1** (Supervised contrastive loss (SC loss)). *Let $Z$ be an $N$-point configuration (assuming that all $z$s are normalized), $Z = (z_1, \ldots, z_N) \in (\mathbb{S}^{h-1})^N$, with labels $Y = (y_1, \ldots, y_N) \in ([K])^N$, and $3 \leq K \leq h + 1$. The supervised contrastive loss $\mathcal{L}_{SC}(\cdot; Y) : (\mathbb{S}^{h-1})^N \to \mathbb{R}$ is defined as*

$$\mathcal{L}_{SC} = \sum_{k=1}^{K} \sum_{i \in B_k} \mathcal{L}_{SC}^{k,i}, \text{ where } \mathcal{L}_{SC}^{k,i} = -\frac{\mathbb{1}_{\{N_k > 1\}}}{N_k - 1} \sum_{j \in B_k \setminus \{i\}} \log \left( \frac{\exp(\langle z_i, z_j \rangle / \tau)}{\sum_{l \in B \setminus \{i\}} \exp(\langle z_i, z_l \rangle / \tau)} \right) \quad (1)$$

**Definition 2** (Equidistant/regular simplex). *Let $h, K \in \mathbb{N}$ and $K \leq h + 1$. A $K$-point configuration $\zeta = (\zeta_1, \ldots, \zeta_K) \in (\mathbb{S}^{h-1})^N$ form the vertices of an equidistant simplex inscribed in the unit hypersphere if and only if the following conditions hold:*

*(1) $\forall i \in [K], \|\zeta_i\| = 1$*

*(2) $\exists d \in \mathbb{R}, \forall i, j$ and $1 \leq i < j \leq K, d = \langle \zeta_i, \zeta_j \rangle$*

*$\zeta$ form the vertices of a regular simplex if and only if (1), (2) and the following condition holds:*

*(3) $\sum_{i \in [K]} \zeta_i = 0$*

## 3.2 OPTIMAL REPRESENTATION CONFIGURATION

In this subsection, we assume a sufficiently powerful encoder capable of realizing any representation configuration, and set the temperature parameter (in Eq. (1)) to $\tau = 1$.

**Optimal Representation Configuration for Balanced Data.** When the training data is balanced, Theorem 1 states that the SC loss attains its minimum if and only if the features of each class converge at their respective class centers, and the centers of all classes form a regular simplex.

**Theorem 1.** *Let $Z$ be an $N$-point configuration (assuming that all $z$s are normalized), $Z = (z_1, \ldots, z_N) \in (\mathbb{S}^{h-1})^N$, with labels $Y = (y_1, \ldots, y_N) \in ([K])^N$, and $3 \leq K \leq h + 1$. When $Y$ is balanced, hence $\forall i \in [K], N_k = \frac{N}{K}$, it holds that:*

$$\mathcal{L}_{SC} \geq N \log \left( \left( \frac{N}{K} - 1 \right) + \frac{N(K-1)}{K} \exp \left( -\frac{K}{K-1} \right) \right) \quad (2)$$

*where equality is attained if and only if there exists a configuration of $\bar{Z} = (\bar{z}_1, \ldots, \bar{z}_K) \in (\mathbb{S}^{h-1})^K$ such that:*

*(A1) $i \in B_k$, $z_i = \bar{z}_k$.*

*(A2) $\bar{Z}$ form a regular simplex inscribed in the unit hypersphere.*

**Remark 1.** *This theorem has been previously established in Graf et al. (2021). In this paper, we provide a refined proof (Appendix B.1) that dose not presume $\mathcal{L}_{\mathrm{SC}}^{k,i}$ (in Eq. (1)) to be the same when $k$ varies, as was done in (S39) of Graf et al. (2021). This allows us to extend the analysis to more general center configurations, particularly laying the foundation for the imbalanced data case (Theorem 2).*

**Optimal Representation Configuration for Imbalanced Data.** When the training data is imbalanced, we first find the tight lower bound function $f$ of $\mathcal{L}_{\mathrm{SC}}$, assuming all features having converged to their respective class centers. $f$ therefore only depends on the center configuration. We then determine the optimal representation configuration by minimizing $f$. In the one-vs-all scenario, Theorem 2 states that the SC loss is minimized if and only if the features of classes 2 to $K$ converge to the vertices of an equidistant simplex while the features of class 1 converge to the point that is perpendicular to the equidistant simplex (more explanations in Appendix A.2)

**Theorem 2.** *Let $Z$ be an $N$-point configuration (assuming that all $z$s are normalized), $Z = (z_1, \ldots, z_N) \in (\mathbb{S}^{h-1})^N$, with labels $Y = (y_1, \ldots, y_N) \in ([K])^N$, and $3 \leq K \leq h + 1$. If $\forall k \in \{2, \ldots, K\}, N_k = a_2 \geq 4$, and $\exists \rho > 0$ such that $N_1 = a_1 = \rho a_2 > 1$, it holds that:*

$$\mathcal{L}_{\mathrm{SC}} \geq f(\cos(\theta_1), \cos(\theta_2)), \tag{3}$$

*where $f(\cdot) : \mathbb{R} \times \mathbb{R} \to \mathbb{R}$ is defined as:*

$$
\begin{aligned}
f(x_1, x_2) = {} & \rho a_2 \log \left( (\rho a_2 - 1) + e^{-1} (K-1) a_2 \exp(x_1) \right) \\
& + (K-1) a_2 \log \left( (a_2 - 1) + e^{-1} \left( (K-2) a_2 \exp(x_2) + \rho a_2 \exp(x_1) \right) \right),
\end{aligned} \tag{4}
$$

*and equality is attained if and only if there exists a configuration of $\bar{Z} = (\bar{z}_1, \ldots, \bar{z}_K) \in (\mathbb{S}^{h-1})^K$ such that:*

*(A3) $i \in B_k$, $z_i = \bar{z}_k$.*

*(A4) $(\bar{z}_2, \ldots, \bar{z}_K)$ form an equidistant simplex whose vertex–center–vertex angle equals $\theta_2$.*

*(A5) $\forall k \in \{2, \ldots, K\}, \langle \bar{z}_1, \bar{z}_k \rangle = \cos(\theta_1)$ and $\cos(\theta_2) = \frac{(K-1)\cos^2(\theta_1) - 1}{K-2}$.*

*The numerical relationship between $\rho$ and $\theta_1$ can be summarized as:*

*(Case 1) $\rho < 1$: $\theta_1 \in \left( \cos^{-1}(-\frac{1}{K-1}), 0 \right)$*

*(Case 2) $\rho = 1$: $\theta_1 = \cos^{-1}(-\frac{1}{K-1})$.*

*(Case 3) $1 < \rho < R(K, a_2)$: $\theta_1 \in \left( -\pi, \cos^{-1}(-\frac{1}{K-1}) \right)$*

*(Case 4) $\rho \geq R(K, a_2)$: $\theta_1 = -\pi$.*

*Let $b_1 = (K-1)(1 + e^{-2} - 2e^2) a_2 - 2$, $b_2 = 8(1 + e^{-2})(K-1) a_2((K-1) a_2 - e^2)$, then $R(K, a_2)$ be defined as:*

$$R(K, a_2) = \frac{-b_1 + \sqrt{b_1^2 + b_2}}{2(1 + e^{-2}) a_2}. \tag{5}$$

Detailed proof is provided in Appendix B.2. We show that $\theta_2$ depends on $\theta_1$ when the representation configuration reaches its optimal. Let $g(\cos(\theta_1)) = f(\cos(\theta_1), \frac{(K-1)\cos^2(\theta_1) - 1}{K-2})$ and we prove that $g$ is a convex function of $\cos(\theta_1)$. Therefore, $g$ has one and only one minimal value within a given domain. We also show that $g'$ is an increasing function of $\rho$. So as $\rho$ increases, $\theta_1$ increases and $\theta_2$ decreases. Here, $\theta_1$ measures the extent of dominance of the head class.

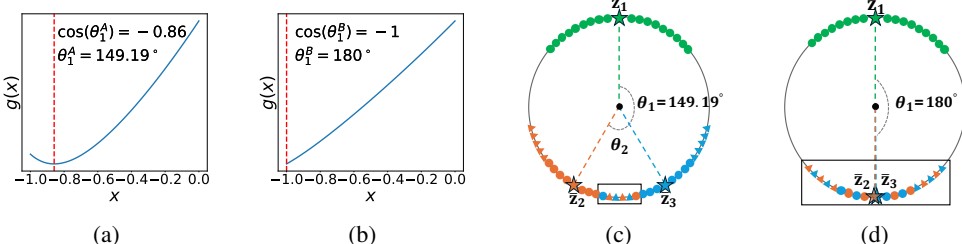

Figure 2: Numerical example. The sample size ratio is **10:1:1** in Example A and **100:1:1** in Example B. (a) $g(x)$ of Example A. (b) $g(x)$ of Example B. (c) Representations of Example A. (d) Representations of Example B. Stars are empirical centers. Circles are available samples, triangles are missing samples. Black boxes are overlapping regions.

**Remark 2.** $R(K, a_2)$ *in Eq.* (5) *can be roughly simplified as a linear function only respect to* $K$:

$$R'(K) = (K - 1)\frac{-(1 + e^{-2} - 2e^2) + \sqrt{(1 + e^{-2} - 2e^2)^2 + 8(1 + e^{-2})}}{2(1 + e^{-2})} \approx 12.16(K - 1) \quad (6)$$

$R'(K)$ *provides an approximate estimate to distinguish Case 3 and Case 4 in Theorem 2.*

**Numerical Examples.** To quantify the extent that long-tailed data skew the feature space, we consider two examples with $K = 3$ classes. Assume that tail classes in both samples have $N_2 = N_3 = 50$ samples. In Example A, set $\rho_A = 10$, and the head class having $N_1^A = 500$ samples. In Example B, set $\rho_B = 100$ and $N_1^B = 5000$. We have $\rho_A < R'(3) < \rho_B$. All samples are mapped to a unit circle ($\mathbb{S}^1$). Then $\theta_1^A = 149.19°$, $\theta_2^A = 61.63°$, $\theta_1^B = 180°$ and $\theta_2^B = 0°$ can be found when $g$ is minimized. Fig. 2 visualizes values of the lower bound function $f$ and the empirical representations of both examples.

## 4 METHOD

### 4.1 CHALLENGES IN LONG-TAILED REPRESENTATION LEARNING

In this subsection, we discuss the challenges faced in long tailed representation learning.

**Skewed Center Configuration.** Theorem 2 reveals that long-tailed data force tail classes' centers to shrink or even collapse. We refer to this phenomenon as the skewed center configuration. This leads to the feature distributions of the tail classes partially (Fig. 2c) or fully (Fig. 2d) overlapping. As a result, samples in the overlapping regions become inseparable and cannot be distinguished by a classifier.

**Distribution Gap.** One may consider rearranging the center configuration back to the symmetrical one to separate the overlapping features. This approach implicitly assumes that the distribution of training data, $\mathcal{P}_{\text{train}}$, is the same with the true distribution of the underlying data, $\mathcal{P}_{\text{true}}$. However, due to the limited sample sizes of tail classes, a discrepancy often exists between $\mathcal{P}_{\text{train}}$ and $\mathcal{P}_{\text{true}}$. We refer to this phenomenon as distribution gap. When it occurs, rearranging the empirical center configuration can separate the training data but cannot ensure the separation of testing data. And doing this may even causes overlapping distributions between head and tail classes (as depicted in Fig. 3).

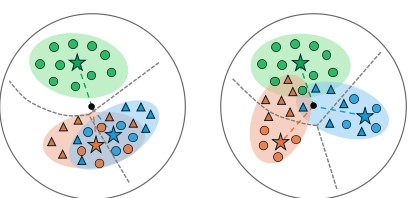

Figure 3: long-tailed data representations. Left: Before center correction. Right: After center correction with distribution shifts. To save space, we defer the legend to Fig. 4.

### 4.2 FEATRECON

To address the problems discussed above, we designed our method to rebalance the skewed feature distributions of long-tailed data to be symmetric and linearly separable.

Figure 4: One iteration of our algorithm. (a) Estimation of the confidence supports with training data. The supports are drawn in dashed magenta circles. (b) Regularization of the support by head class statistics. (c) Generating synthetic features to fill the supports (cross markers). (d) Optimization of the SC loss separates the tail classes and their supports.

Theorem 2 suggests that balancing the sample sizes across all classes can correct the center configuration. To achieve this, we directly generate synthetic features in the feature space. Since all features are normalized (i.e, $Z \in \mathbb{S}^{h-1}$), it is reasonable to assume that the features of each class fall within a hyperspherical cap on $\mathbb{S}^{h-1}$, parameterized by a center and a "radius" – a fixed central angle. Thus, for each class, we estimate its confidence support as a hyperspherical cap that contains the majority of features. We then sample synthetic features from these supports. Each support is filled with real and synthetic features, and features from nearby classes are pushed away as training progresses.

However, since tail classes have limited sample sizes, their estimated supports are unreliable. To prevent missing features of tail classes from falling outside their respective supports and overlapping with the features of the head classes, we regularize tail classes' estimation with the statistics of neighboring head classes. Since synthetic samples encourage the learned representations to be symmetrical, as long as the central angle of any confidence support is sufficiently small (at most $\frac{1}{2}\cos^{-1}(-\frac{1}{K-1})$), these confidence supports are guaranteed to be linearly separable when training reaches the optimal.

The procedure is illustrated in Fig. 4. In Fig. 4a, we estimate the confidence support based on limited training samples. In Fig. 4b, the confidence supports of tail classes are regularized using the head class statistics. In Fig. 4c, we generate synthetic samples that fill these confidence supports. Finally, by minimizing the SC loss, class centers are moved to equal distance from each other, and the supports are guaranteed to be linearly separable (Fig. 4d). In practice, we repeat the procedure iteratively.

**Confidence Supports Estimation.** A hyperspherical cap can be characterized by its center and a central angle. For class $k$, we estimate these parameters as follows:

$$\hat{\mu}_k = \frac{1}{N_k} \sum_{i \in B_k} z_i, \text{ and } \hat{\theta}_k = Q_\alpha \{\cos^{-1}(z_i \cdot \frac{\hat{\mu}_k}{||\hat{\mu}_k||})|y_i = k\}, \tag{7}$$

where $Q_\alpha$ denotes the $\alpha$ quantile.

**Head Class Regularization.** For tail classes, the statistics are regularized using the statistics of $\mathcal{P}_{\text{true}}$ from head classes, which are estimated more accurately due to sufficient training samples. This improves the robustness of tail class parameter estimation. Specifically, for a tail class $k$, we select the top $q$ head classes ($\mathcal{C}_h$) with the highest similarities to its class center $\hat{\mu}$:

$$\mathcal{C}_k^q = \left\{ i \mid \hat{\mu}_i \cdot \hat{\mu}_k \in \text{top}_q\left(\mathcal{S}_k\right)\right\}, \text{ where } \mathcal{S}_k = \{\hat{\mu}_i \cdot \hat{\mu}_k \mid i \in \mathcal{C}_h\}, \tag{8}$$

and regularize its statistics using those from the selected head classes $\mathcal{C}_k^q$ as follows:

$$\hat{\mu}_k' = (1-\gamma) \sum \omega_k^c \hat{\mu}_c + \gamma \hat{\mu}_k \text{ and } \hat{\theta}_k' = (1-\gamma) \sum \omega_k^c \hat{\theta}_c + \gamma \hat{\theta}_k \tag{9}$$

where $\omega_k^c = \frac{N_k \hat{\mu}_i \cdot \hat{\mu}_k}{\sum_{j \in \mathcal{C}_k^q} N_j \hat{\mu}_i \cdot \hat{\mu}_j}$ is the regularization weight, and $\gamma$ is the regularization magnitude.

**Feature Generation.** The estimated confidence support of class $k$ is defined as:

$$\tilde{\mathcal{Z}}_k = \left\{ \tilde{z} \in \mathbb{S}^{h-1} \mid \tilde{z}^\top \frac{\hat{\mu}_k'}{||\hat{\mu}_k'||} \geq \cos(\bar{\theta}_k) \right\}, \text{ where } \bar{\theta}_k = \min\{\hat{\theta}_k', \frac{1}{2}\cos^{-1}(-\frac{1}{K-1})\}. \tag{10}$$

Table 1: Top-1 accuracy of ResNet-32 on CIFAR-10/100-LT datasets with different imbalance factors.

| Dataset | CIFAR-10-LT | | | CIFAR-100-LT | | |
|---|---|---|---|---|---|---|
| Imbalance Ratio ($\rho$) | 100 | 50 | 10 | 100 | 50 | 10 |
| CE | 70.4 | 74.8 | 86.4 | 38.3 | 43.9 | 55.7 |
| Focal Loss (Lin et al., 2017) | 70.4 | 76.7 | 86.7 | 38.4 | 44.3 | 55.8 |
| CB-Focal (Cui et al., 2019) | 74.6 | 79.3 | 87.1 | 39.6 | 45.2 | 58.0 |
| LDAM-DRW (Cao et al., 2019) | 77.0 | 81.0 | 88.2 | 42.0 | 46.6 | 58.7 |
| CB-DA-LDAM (Jamal et al., 2020) | 80.0 | 82.2 | 87.4 | 44.1 | 49.2 | 58.0 |
| CE-OTmix (Gao et al., 2024) | 78.3 | 83.4 | 90.2 | 46.4 | 40.7 | 61.6 |
| DWR-OTmix (Cao et al., 2019; Gao et al., 2024) | 83.1 | 86.4 | 90.6 | 48.0 | 52.60 | 62.7 |
| SCL (Khosla et al., 2020) - | - | - | - | 42.1 | 45.2 | 54.8 |
| Hybrid-SC (Wang et al., 2021) | 81.4 | 85.4 | 91.1 | 46.7 | 51.9 | 63.5 |
| Hybrid-PSC (Wang et al., 2021) | 78.8 | 83.9 | 91.0 | 45.0 | 48.9 | 62.7 |
| KCL (Kang et al., 2021) | 77.6 | 81.7 | 88.0 | 42.8 | 46.3 | 57.6 |
| TSC (Li et al., 2022) | 79.7 | 82.9 | 88.7 | 43.8 | 47.4 | 59.0 |
| BCL (Zhu et al., 2022) | 84.3 | 87.2 | 91.1 | 51.8 | 56.6 | 64.9 |
| SBCL (Hou et al., 2023) - | - | - | - | 44.9 | 48.7 | 57.9 |
| FeatRecon | **85.2** | **87.8** | **91.6** | **52.5** | **57.0** | **65.3** |

And we sample $N_{gen}$ features for each class from the respective confidence support.

**Temperature Adjustment.** Previous works (Wang & Liu, 2021) have revealed how temperature parameter $\tau$ affects the representation learning. A larger $\tau$ places more emphasis on inter-class separation, allowing head classes, which have more samples, to learn more accurate class boundaries where tail classes should remain distant. Inspired by this, we adjust the temperature for class $k$ as:

$$\tau_k = \left(1 - 0.5\left(1 + \cos\left(\pi \frac{N_k - N_{\min}}{N_{\max} - N_{\min}}\right)\right)\right) \times (\tau_+ - \tau_-) + \tau_- \tag{11}$$

where $\tau_+, \tau_-$ denote the upper and lower bounds of $\tau$, respectively. Also, Khosla et al. (2020) shows that $\tau$ also controls the gradient scale: the larger $\tau$, the smaller the gradient. Therefore, we rebalance the gradient scale of the samples by adjusting the weight of $\mathcal{L}_{SC}^{k,i}$ with a scalar parameter $\frac{\tau_k}{\tau_-}$.

**Training Framework.** Our training framework follows (Zhu et al., 2022). The model consists of: 1) a base encoder $f : X \rightarrow h$ that extracts latent embeddings; 2) a prediction head $l : h \rightarrow p$ that produces model predictions $p = l \circ f(\mathcal{X})$; 3) a projection head $g : h \rightarrow z$ that generates normalized representations $z = g \circ f(\mathcal{X})$.

The prediction head is optimized using the training data with cross entropy loss and logit compensation (Menon et al., 2021). Let $\mathbb{P}(y)$ be class priors and $\delta_y = \log \mathbb{P}_y$. Then the $\mathcal{L}_x$ is:

$$\mathcal{L}_x(y, l \circ f(x)) = -\log \frac{\exp(p_y + \delta_y)}{\sum_{y' \in [\mathcal{Y}]} \exp(p_{y'} + \delta_{y'})} \tag{12}$$

The projection head is optimized using both real and synthetic features with supervised contrastive loss $\mathcal{L}_{SC}$. The final objective is:

$$\mathcal{L} = \lambda_x \mathcal{L}_x + \lambda_c \mathcal{L}_{SC} \tag{13}$$

where $\lambda_x$ and $\lambda_c$ are hyperparameters that control relative strength among different losses.

## 5 EXPERIMENTS

### 5.1 DATASET AND IMPLEMENTATION DETAILS.

**Dataset. CIFAR-10-LT and CIFAR-100-LT** are the imbalanced subsets of the original CIFAR-10 and CIFAR-100 (Krizhevsky et al., 2009), following (Kang et al., 2021; Li et al., 2022; Zhu et al., 2022). We set the imbalance factor $\rho = N_{max}/N_{min}$ to be 100, 50, and 10.

**ImageNet-LT** (Liu et al., 2019) is the subset of the original ImageNet (Deng et al., 2009), with the training set sampled with a Pareto distribution with power value $\alpha = 0.6$ and testing set unchanged. The imbalance factor is 256, with the most frequent class having 1280 samples and the least frequent one having 5 samples.

**iNaturalist 2018** (Van Horn et al., 2018) is a large-scale long-tailed dataset that contains 437.5K images from 8,142 classes with an extremely imbalanced distribution.

Following (Zhu et al., 2022), we train our model on the long-tailed training sets and evaluate on the balanced testing sets. We divide the testing sets into three subsets: many (with more than 100 instances), medium (with 20 to 100 instances), and few (with less than 20 instances) splits.

Table 2: Top-1 accuracy of ResNet-32 on CIFAR-100-LT with imbalance factor equaling 100.

| | Methods | Many | Medium | Few | All |
|---|---|---|---|---|---|
| 200 epochs | Hybrid-SC (Wang et al., 2021) | - | - | - | 46.7 |
| | DRO-LT (Samuel & Chechik, 2021) | 64.7 | 50.0 | 23.8 | 47.3 |
| | RIDE(3 experts) (Wang et al., 2020) | 68.1 | 49.2 | 23.9 | 48.0 |
| | BCL (Zhu et al., 2022) | 67.0 | 52.9 | 33.2 | 51.8 |
| | FeatRecon (Ours) | **68.2** | **53.3** | **34.0** | **52.5** |
| 400 epochs | Balanced Softmax (Ren et al., 2020) | - | - | - | 50.8 |
| | PaCo (Cui et al., 2021) | - | - | - | 52.0 |
| | GPaCo (Cui et al., 2023) | - | - | - | 52.3 |
| | BCL (Zhu et al., 2022) | 69.6 | 53.8 | 33.3 | 53.4 |
| | FeatRecon (Ours) | **70.0** | **53.9** | **35.5** | **54.0** |

**Implementation Details.** Our implementation follows (Zhu et al., 2022). For both CIFAR-10-LT and CIFAR-100-LT, we adopt the ResNet-32 as the backbone. The projection head is a 2-layer MLP that generates 128-dimensional embeddings. Dimension of the hidden layer is 512. Our model is trained for 200 epochs with a batch size of 256 and with a SGD optimizer. The momentum is 0.9 and the weight decay is $5e^{-4}$. The learning rate warms up 0.15 in the first 5 epochs and decay by 0.1 at the 160th and 180th epochs. For data augmentation, we adopt AutoAug (Cubuk et al., 2019) and Cutout (DeVries & Taylor, 2017) for the classification head, and adopt SimAug (Chen et al., 2020) for the projection head. For hyperparameters, we set $\lambda_x = 2.0$, $\lambda_c = 0.6$, $\alpha = 0.99$, and $\tau_- = 0.1$, $\tau_+$ is scheduled by training epoch between 0 and 1 using method in Kukleva et al. (2023). We also train our model for 400 epochs for finer comparisons on CIFAR-100-LT. In this case, the learning rate warms up in the first 10 epochs and decay at the 360th and 380th epochs.

We adopt ResNet-50 (He et al., 2016) as the model backbone for both ImageNet-LT and iNaturalist 2018. The projection head is a 2-layer MLP that generates 1024-dimensional embeddings. Dimension of the hidden layer is 2048. For data augmentation, we switch the strategy for the projection head to RandAug (Cubuk et al., 2020). Our model is trained for 90 epochs for ImageNet-LT and 100 for iNaturalist 2018 epochs with a batch size of 256 and with a SGD optimizer. The momentum is 0.9 and the weight decay is $5e^{-4}$ for ImageNet-LT and $1e^{-4}$ for iNaturalist 2018. The learning rate is 0.1 for ImageNet-LT and 0.2 for iNaturalist 2018 with a cosine scheduler. Additionally, we train our model for 90 epochs using ResNeXt-50-32x4d (Xie et al., 2017) as the backbone. For hyperparameters, we set $\lambda_x = 1$, $\lambda_c = 0.35$, $\alpha = 0.99$, and $\tau_- = 0.07$, $\tau_+$ is scheduled by training epoch between 0.07 and 1 using method in Kukleva et al. (2023)..

Table 3: Top-1 accuracy of ResNet-50 on ImageNet-LT dataset and iNaturalist 2018 dataset.

| Methods | ImageNet-LT | iNaturalist 2018 |
|---|---|---|
| CE | 41.6 | 61.7 |
| Focal Loss (Lin et al., 2017) | 43.7 | 61.3 |
| LDAM-DRW (Cao et al., 2019) | 49.8 | 64.6 |
| cRT (Kang et al., 2020) | 47.3 | 65.2 |
| $\tau$-norm (Kang et al., 2020) 46.7 | 65.6 | |
| LWS (Kang et al., 2020) 47.7 | 65.9 | |
| Area (Chen et al., 2023) | 49.5 | 68.4 |
| CE-OTmix (Gao et al., 2024) | 52.0 | 69.5 |
| DRW-OTmix (Gao et al., 2024) | 53.4 | 71.1 |
| IWB (Dang et al., 2024) | 55.2 | 71.5 |
| SCL (Khosla et al., 2020) | 49.8 | 66.4 |
| KCL (Kang et al., 2021) | 51.5 | 68.6 |
| TSC (Li et al., 2022) | 52.4 | 69.7 |
| BCL (Zhu et al., 2022) | 56.0 | 71.8 |
| SBCL (Hou et al., 2023) | 53.4 | 70.8 |
| FeatRecon | **56.8** | **72.9** |

## 5.2 RESULTS

**CIFAR-LT** Tab. 1 shows experiment results on CIFAR-10/100-LT datasets with imbalance factor varying among 10, 50, and 100. For baselines, we select methods that only work with classifiers (Lin et al., 2017; Cui et al., 2019; Cao et al., 2019; Jamal et al., 2020; Gao et al., 2024) and methods that work with both representations and classifiers (Khosla et al., 2020; Wang et al., 2021; Kang et al., 2021; Li et al., 2022; Zhu et al., 2022; Hou et al., 2023). We can see that FeatRecon outperforms

baseline models in all settings. Moreover, our model achieves larger performance gain as the imbalance factor increases, proving the effectiveness of our method for long-tailed data. Additionally, in Tab. 2, we provide shot-wise results on CIFAR-100-LT data with imbalance factor of 100. The model is trained for both 200 epochs and 400 epochs for fair comparisons with baselines that are trained under different settings. The results demonstrate the superiority of our approach, especially for the few-shot classes.

**ImageNet-LT** Tab. 3 shows the results of the experiments on the ImageNet-LT dataset using ResNet-50 as the model backbone. Tab. 4 shows the results of the experiments using ResNeXt-50 as the model backbone. We report the overall Top-1 accuracy as well as the Top-1 accuracy of Many-shot, Medium-shot, and Few-shot classes. Similarly to the experiments on CIFAR-LT, we select methods that only work with classifiers (Lin et al., 2017; Cao et al., 2019; Kang et al., 2020; Chen et al., 2023; Gao et al., 2024; Dang et al., 2024) and methods that work with both representations and classifiers (Khosla et al., 2020; Kang et al., 2021; Li et al., 2022; Zhu et al., 2022; Hou et al., 2023) for baselines. The results show that our method outperforms baselines in terms of overall accuracy, demonstrating the effectiveness of our approach for learning classes with synthetic features.

Table 4: Top-1 accuracy of ResNeXt-50 on ImageNet-LT dataset.

| Method | Many | Med | Few | All |
|---|---|---|---|---|
| Focal Loss (Lin et al., 2017) | 64.3 | 37.1 | 8.2 | 43.7 |
| $\tau$-norm (Kang et al., 2020) | 59.1 | 46.9 | 30.7 | 49.4 |
| LWS (Kang et al., 2020) | 60.2 | 47.2 | 30.3 | 49.9 |
| IWB (Dang et al., 2024) | 64.2 | 52.2 | **40.2** | 55.2 |
| BCL (Zhu et al., 2022) | 67.2 | 53.9 | 36.5 | 56.7 |
| Ours | **67.9** | **54.7** | 37.8 | **57.5** |

**iNaturalist 2018** Tab. 3 also lists the experiment results on iNaturalist 2018 dataset. Similarly to results on ImageNet-LT, our method outperforms baselines on the accuracy of tail classes and overall accuracy, highlighting our model's capability of learning from a few samples.

## 5.3 ABLATION STUDY

We evaluate the design of FeatRecon through an ablation study on CIFAR-100-LT dataset, with an imbalance factor of 100. Each model runs for 200 epochs. The results are displayed in Tab. 5. Exp. 1 provides the baseline by training a classifier with logit compensation (LP) (Menon et al., 2021). Exp. 2 introduces an additional projection head and trains feature representations with the SC loss (Khosla et al., 2020). This design brings a $0.9\%$ performance improvement, underscoring the benefit of representation learning. In Exp. 3, we balance the sample size across different classes by naively upsampling (Up Sam) the existing features for representation learning. However, this approach has no significantly positive effect. It highlights the effectiveness of our synthetic feature generation method (Feat Gen), shown in Exp. 4, which brings a $2.0\%$ performance gain. In Exp. 5, we validate the benefit of training with temperature adjustment (Temp Adj), which leads to a total $2.3\%$ performance increase.

Table 5: Ablating model components.

| Exp | LC | SC | Up Sam | Feat Gen | Temp Adj | Accuracy | $\Delta$ |
|---|---|---|---|---|---|---|---|
| 1 | ✓ | | | | | 50.2 | |
| 2 | ✓ | ✓ | | | | 51.1 | +0.9 |
| 3 | ✓ | ✓ | ✓ | | | 51.2 | +1.0 |
| 4 | ✓ | ✓ | | ✓ | | 52.2 | +2.0 |
| 5 | ✓ | ✓ | | ✓ | ✓ | **52.5** | **+2.3** |

## 6 CONCLUSION

In this paper, we establish a theoretical framework to investigate the optimal representation configuration for long-tailed data. We prove that centers of tail classes either shrink or collapse. Building on this analysis, we explore the challenges associated with long-tailed representations, identifying two key issues: skewed center configurations and distribution shifts. Inspired by our findings, we introduce a novel method for long-tailed representation learning. Our method reconstructs the feature space for long-tailed data, arranging the representations of each class into symmetric and linearly separable regions. We demonstrate the effectiveness of our approach on several benchmark datasets and the results show that our method achieves state-of-the-art performance.

ACKNOWLEDGEMENTS

This research was partially supported by the National Science Foundation (NSF) grant CCF-2144901, CPS-2128350, the National Institute of Health (NIH) Grants R01NS143143, R01CA297843, and the Stony Brook Trustees Faculty Award. Lingjie Yi is also supported in part by Bloomberg Data Science Fellowship.

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

# A  APPENDIX A

## A.1  PSEUDO ALGORITHMS

### A.1.1  TRAINING PROCESS

In this section, we first present the training procedure of FeatRecon. FeatRecon is a heuristic method that iteratively generates synthetic features, adjust representations, and re-estimate confidence supports at each step of the training process.

---

**Algorithm 1:** FeatRecon Algorithm

**Input:** Available training samples $\mathcal{D} = \{x_i, y_i\}_{i \in B_k}$ from $K$ classes, the quantile parameter $\alpha$,
the regularization magnitude $\gamma$ and $m$ which controls the total number of synthetic features.

1 **for** $t = 1, \ldots, T$ **do**
2      Sample a mini-batch $\{\boldsymbol{x}_i, y_i\}_{i=1}^{B}$ from $\mathcal{D}$ ;
3      **for** $k = 1, \ldots, K$ **do**
4          Estimate batch-wise support centers and update with epoch-wise estimation ;
5          **if** *class k is a tail class* **then**
6              Regularize all statistics as Eq. (8) and Eq. (9);
7          **end**
8          Generate $N_{\text{gen}}$ synthetic features for class $k$ ;
9      **end**
10      Compute the cross entropy loss $\mathcal{L}_x$ (Eq. (12)) with training data ;
11      Compute the supervised contrastive loss $\mathcal{L}_{SC}$ (Eq. (1));
12      Update model with loss $\mathcal{L} = \lambda_x \mathcal{L}_x + \lambda_c \mathcal{L}_{SC}$ ;
13      Estimate confident supports as Eq. (7) for all classes
14 **end**
15 **Return** Trained model

---

### A.1.2  SYNTHETIC FEATURE GENERATION

Next, we present the pseudo-algorithms of synthetic feature generation. This part follows the core idea of Su (2021). Suppose that the angles between the samples of class $k$ and their respective center $\mu$ follow a given distribution $p_k(\theta; \mu)$. To generate synthetic features within the confidence support of the class $k$, we first sample angles from $p_k(\theta; \tilde{\mu}'_k)$ but only retain samples that satisfy $0 \leq \theta \leq \bar{\theta}_k$. So we have:

$$\Theta_k = \{\theta \sim p_k(\theta; \tilde{\mu}'_k); \theta < \bar{\theta}_k\} \tag{14}$$

Then we can generate samples whose angle to the center is $\theta$ via:

$$x = \cos(\theta) \times \tilde{\mu}'_k + \sin(\theta) \times \nu$$
$$\text{where } \nu \sim U(\mathcal{S}^{h-2}) \text{ and } \nu \cdot \tilde{\mu}'_k = 0 \tag{15}$$

Here, $\nu$ are sampled from the uniform distribution on the surface of a $h - 2$ hypersphere $\mathcal{S}^{h-2}$ that is perpendicular to $\tilde{\mu}'_k$. And we can sample $\nu$ via

$$s \sim \mathcal{N}(0, I_h)$$
$$\nu = \frac{s - \langle s, \tilde{\mu}'_k \rangle \times \tilde{\mu}'_k}{\|s - \langle s, \tilde{\mu}'_k \rangle \times \tilde{\mu}'_k\|} \tag{16}$$

And we can verify that the samples generated via the above method fall within the confidence support and their angles to the class centers follow a truncated version of $p_k(\theta; \tilde{\mu}'_k)$.

The entire procedure can be summarized by Algorithm. 2

---

**Algorithm 2:** Synthetic Feature Generation Algorithm

---

**Input:** Parameters of the estimated confidence support $\{\hat{\mu}'_k, \bar{\theta}'_k\}$, angle distributions $p_k(\theta; \tilde{\mu}'_k)$, numbers of generated samples $G_k$, where $k \in [K]$

1 **for** $k = 1, \ldots, K$ **do**
2     **Sample** $N_k$ iid. $\{\theta_k\}$ as described in Eq. (14) ;
3     **Sample** $N_k$ iid. $\{\nu_k\}$ as described in Eq. (16) ;
4     **Calculate** $\{x_k\}$ as described in Eq. (15) ;
5 **end**
6 **Return** $\{x_k\}_{k \in [K]}$

---

In terms of generating synthetic features, we sample $\theta$ from the normal distribution. This evolves to estimate two more parameters, the center and the variance of $\theta$. We estimate both parameters once every epoch and regularize them at each mini-batch.

## A.2 MORE EXPLANATION OF THEOREM 2

In this subsection, we provide a more detailed mathematical explanation with respect to Theorem 2.

It states the necessary and sufficient conditions on the representation configuration for the SC loss to attain its minimal. (A3) states thyat representations of each class converge to the respective class centers. (A4) states that the centers of class 2 to $K$ form a $K - 2$ equidistant simplex, the angles between whose vertices are all equal $\theta_2$. (A5) states that the vector between the spherical center and the center of class 1 is perpendicular to the equidistant simplex, and the angle between class 1's center and other classes' center all equal $\theta_1$. And $\cos(\theta_2) = \frac{(K-1)\cos^2(\theta_1)-1}{K-2}$.

The additional numerical relationship describes that the entire dynamic of the configuration of all centers as $\rho$ increases from $0$ to $+\infty$. More specifically:

(Case 1)   $0 < \rho < 1$: $\frac{\pi}{2} < \theta_1 < \cos^{-1}(-\frac{1}{K-1}) < \theta_2 < \cos^{-1}(-\frac{1}{K-2})$

(Case 2)   $\rho = 1$: it becomes a data balance case where $\theta_1 = \theta_2 = \cos^{-1}(-\frac{1}{K-1})$. This indicates that all class centers form a $K - 1$ regular simplex.

(Case 3)   $1 < \rho < R(K, a_2)$: as $\rho$ continues to increase, it becomes a long-tailed problem. The head class ($1^{st}$) increasingly dominates the feature space as $\pi > \theta_1 > \cos^{-1}(-\frac{1}{K-1}) > \theta_2 > 0$. At this stage, centers of tail classes increasingly shrink together.

(Case 4)   $\rho > R(K, a_2)$: centers of tail classes collapses with $\theta_2 = 0$ and $\theta_1 = \pi$

In both long-tailed cases, $\theta_1$ measures the extent that a head class dominate the feature space. Also, Theorem 1 is a special case of Theorem 2 (Case 2).

## A.3 LIMITATIONS

While our theoretical framework opens a door to study long-tailed representation, it's currently limited to the simple one vs. all case. The solution for more general cases remains unsolved.

To solve this problem, it is crucial to identify the numerical relationships among the angles between different class centers. As suggested in Lemma 5, if there are K classes with M distinct sample sizes, there will be $\binom{M}{2}$ distinct angles. Determining the numerical relationships among these angles poses a significant challenge.

However, while our theory primarily addresses the one-vs-all scenario, it unveils a crucial insight into how the head classes dominate the feature space and provides guidance on algorithm design.

## A.4 REPRESENTATION VISUALIZATION

In Fig. 5, we visualize the learned representations of CIFAR-10 testing images with imbalance factor equaling 100 using $t$-SNE (Van Der Maaten & Hinton, 2008). The results show that with the help of synthetic features generated by FeatRecon, the resulting testing distributions of different classes are more separated.

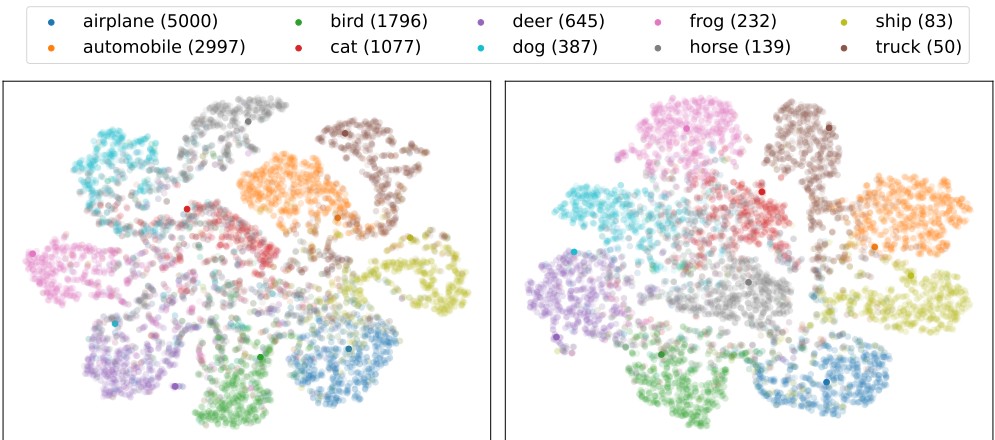

Figure 5: $t$-SNE visualization of CIFAR-10 testing set. (left) Learned representations without synthetic features. (Right) Learned representations with synthetic features generated by FeatRecon. Numbers in the legend after class names represents the numbers of training samples from this class.

# B    APPENDIX B

## B.1    PROOF OF THEOREM 1

In this section, we provide proofs of Theorem 1 proposed in Sec. 3.2. Our proof is different from what is shown in (Graf et al., 2021; Zhu et al., 2022) in order to take into account the long-tailed distribution. For convenience in reading, let us recall some related notions and definitions.

- $h, N, K \in \mathbb{N}$
- $\mathcal{Z} = \mathbb{R}^h$
- $\mathbb{S}^{h-1} = \left\{ z \in \mathbb{R}^h : \|z\| = 1 \right\}$
- $\mathcal{Y} = \{1, \dots, K\} = [K]$
- $B = \{1, \dots, N\} = [N]$
- $B_k = \{i : i \in B, y_i = k\}$
- $N_k = |B_k|$

**Definition 1** (Supervised contrastive loss) Let $Z$ be an $N$-point configuration (assuming that all $zs$ are normalized), $Z = (z_1, \dots, z_N) \in (\mathbb{S}^{h-1})^N$, with labels $Y = (y_1, \dots, y_N) \in ([K])^N$, and $K \leq h + 1$. Let $B = [N]$, $B_k = \{i : i \in B, y_i = k\}$ and $N_k = |B_k|$. The supervised contrastive loss $\mathcal{L}_{\mathrm{SC}}(\cdot; Y) : (\mathbb{S}^{h-1})^N \to \mathbb{R}$ is defined as:

$$\mathcal{L}_{\mathrm{SC}} = \sum_{k=1}^{K} \sum_{i \in B_k} \mathcal{L}_{\mathrm{SC}}^{k,i}, \text{ where } \mathcal{L}_{\mathrm{SC}}^i = -\frac{\mathbb{1}_{\{N_k > 1\}}}{N_k - 1} \sum_{j \in B_k \setminus \{i\}} \log \left( \frac{\exp\left(\langle z_i, z_j \rangle\right)}{\sum_{l \in B \setminus \{i\}} \exp\left(\langle z_i, z_l \rangle\right)} \right).$$

**Definition 3** (Equidistant/regular Simplex) Let $h, K \in \mathbb{N}$ with $K \leq h + 1$. A K-point configuration $\zeta = (\zeta_1, \dots, \zeta_K) \in (\mathbb{S}^{h-1})^N$ forms the vertices of an equidistant simplex inscribed in the unit hypersphere if and only if all of the following conditions hold:

(1) $\forall i \in [K], \|\zeta_i\| = 1$

(2) $\exists d \in \mathbb{R}, \forall i, j \text{ and } 1 \leq i < j \leq K, d = \langle \zeta_i, \zeta_j \rangle$

And $\zeta$ form the vertices of a regular simplex inscribed in the unit hypersphere if and only if (1), (2) and the following condition holds:

(3) $\sum_{i \in [K]} \zeta_i = 0$

**Theorem 1** Let $Z$ be a $N$-point configuration (assuming that all $zs$ are normalized), $Z = (z_1, \dots, z_N) \in (\mathbb{S}^{h-1})^N$, with labels $Y = (y_1, \dots, y_N) \in ([K])^N$, and $K \leq h + 1$. Let $B = [N]$, $B_k = \{i : i \in B, y_i = k\}$ and $N_k = |B_k|$. When $Y$ is balanced, hence $\forall i \in [K], N_k = \frac{N}{K}$, it holds that:

$$\mathcal{L}_{\mathrm{SC}} \geq N \log \left( \left( \frac{N}{K} - 1 \right) + \frac{N(K-1)}{K} \exp \left( -\frac{K}{K-1} \right) \right),$$

where equality is attained if and only if there exists a configuration of $\bar{Z} = (\bar{z}_1, \dots, \bar{z}_K) \in (\mathbb{S}^{h-1})^K$ such that:

(A1) $i \in B_k, z_i = \bar{z}_k$.

(A2) $\bar{Z}$ form a regular simplex inscribed in the unit-hyperspher.

### B.1.1 STEPS OF PROOF

First let's rewrite $\mathcal{L}_{\mathrm{SC}}^{k,i}$ and $\mathcal{L}_{\mathrm{SC}}$ (assuming $\forall k \in [K], N_k > 1$).

$$
\begin{aligned}
\mathcal{L}_{\mathrm{SC}}^{k,i} &= -\frac{1}{N_k - 1} \sum_{j \in B_k \setminus \{i\}} \log \left( \frac{\exp(\langle z_i, z_j \rangle)}{\sum_{l \in B \setminus \{i\}} \exp(\langle z_i, z_l \rangle)} \right) \\
&= \frac{1}{N_k - 1} \sum_{j \in B_k \setminus \{i\}} \log \left( \frac{\sum_{l \in B \setminus \{i\}} \exp(\langle z_i, z_l \rangle)}{\exp(\langle z_i, z_j \rangle)} \right) \\
&= \frac{1}{N_k - 1} \log \left( \frac{\left( \sum_{l \in B \setminus \{i\}} \exp(\langle z_i, z_l \rangle) \right)^{N_k - 1}}{\prod_{j \in B_k \setminus \{i\}} \exp(\langle z_i, z_j \rangle)} \right) \\
&= \log \left( \frac{\sum_{l \in B \setminus \{i\}} \exp(\langle z_i, z_l \rangle)}{\exp \left( \sum_{j \in B_k \setminus \{i\}} \langle z_i, z_j \rangle \right)^{\frac{1}{N_k - 1}}} \right) \\
&= \log \left( \frac{\sum_{l \in B \setminus \{i\}} \exp(\langle z_i, z_l \rangle)}{\exp \left( \frac{1}{N_k - 1} \sum_{j \in B_k \setminus \{i\}} \langle z_i, z_j \rangle \right)} \right),
\end{aligned}
\tag{17}
$$

and hence

$$
\begin{aligned}
\mathcal{L}_{\mathrm{SC}} &= \sum_{k=1}^{K} \sum_{i \in B_k} \mathcal{L}_{\mathrm{SC}}^{k,i} \\
&\overset{Lemma\ 2}{\geq} \sum_{k=1}^{K} N_k \log \left( (N_k - 1) + e^{-1} \sum_{\substack{k' \in [K] \\ k' \neq k}} N_{k'} \exp(\langle \bar{z}_k, \bar{z}_{k'} \rangle) \right),
\end{aligned}
\tag{18}
$$

where $\bar{z}_k = \frac{1}{N_k} \sum_{i \in B_k} z_i$. When $Y$ is balanced, $\forall i \in [K]$, $N_k = \frac{N}{K}$, then

$$
\begin{aligned}
\mathcal{L}_{\mathrm{SC}} &\geq \sum_{k=1}^{K} \frac{N}{K} \log \left( \left( \frac{N}{K} - 1 \right) + e^{-1} \frac{N}{K} \sum_{\substack{k' \in [K] \\ k' \neq k}} \exp(\langle \bar{z}_k, \bar{z}_{k'} \rangle) \right) \\
&\overset{Lemma\ 3}{\geq} N \log \left( \left( \frac{N}{K} - 1 \right) + e^{-1} \frac{N(K-1)}{K} \exp(\beta) \right),
\end{aligned}
\tag{19}
$$

and equality is attained if and only if all of the following conditions hold:

(B1) $\forall i \in B_k$, $z_i = \bar{z}_k$.

(B2) $\forall k \in [K]$ and $k' \in [K] \setminus \{k\}$, $\langle \bar{z}_k, \bar{z}_{k'} \rangle = \beta$.

(B3) There exists a configuration of $\bar{Z} = (\bar{z}_1, \dots, \bar{z}_K)$ such that (B2) holds.

(Case 1) $K = h + 1$: $\beta = -\frac{1}{K-1}$ or $\beta = 1$

(Case 2) $K < h + 1$: $-\frac{1}{K-1} \leq \beta \leq 1$

When $a, b > 0$, $f(x) = \log(a + be^x)$ is a strictly increasing function. And Eq. (19) suggests that the lower bound of $\mathcal{L}_{\mathrm{SC}}$ is a strictly increasing function of $\beta$. When $\beta$ reaches its minimal value, so does $\mathcal{L}_{\mathrm{SC}}$. When $K \leq h + 1$, $\beta_{\min} = -\frac{1}{K-1}$, we have:

$$\mathcal{L}_{\text{SC}} \geq N \log \left( \left( \frac{N}{K} - 1 \right) + e^{-1} \frac{N(K-1)}{K} \exp \left( -\frac{1}{K-1} \right) \right)$$

$$= N \log \left( \left( \frac{N}{K} - 1 \right) + \frac{N(K-1)}{K} \exp \left( -\frac{K}{K-1} \right) \right). \tag{20}$$

When $\beta = -\frac{1}{K-1}$, Lemma 1 shows that (B2) and (B3) imply $\bar{Z} = (\bar{z}_1, \ldots, \bar{z}_K)$ to form a regular simplex. Thus, the conditions for equality can be summarized as: there exists a configuration of $\bar{Z} = (\bar{z}_1, \ldots, \bar{z}_K) \in (\mathbb{S}^{h-1})^K$ such that:

(A1) $i \in B_k, z_i = \bar{z}_k$.

(A2) $\bar{Z}$ form a regular simplex inscribed in the unit hypersphere.

### B.1.2 Lemmas Part 1

In this section, we provide definitions and proofs of lemmas that are used for the proof of Theorem 1.

**Lemma 1.** *Let $Z$ be an $K$ point configuration (assuming that all $z$s are normalized), $Z = (z_1, \ldots, z_K) \in (\mathbb{S}^{h-1})^K$. If $\exists \beta \in \mathbb{R}, \forall i, j \in [K]$ and $i \neq j$ such that all inner products $\langle z_i, z_j \rangle = \beta$ are equal, then one of the following cases holds:*

*(Case 1)* $K > h + 1$: $\beta = 1$.

*(Case 2)* $K = h + 1$: $\beta = -\frac{1}{N-1}$ *or* $\beta = 1$.

*(Case 3)* $K < h + 1$: $-\frac{1}{N-1} \leq \beta \leq 1$.

*And when $\beta = -\frac{1}{K-1}$, $Z = (z_1, \ldots, z_K)$ forms a regular simplex.*

*Proof.* As explained in (Delsarte et al., 1977), there are at most $h + 1$ equidistant points on $\mathbb{S}^{h-1}$ (the size of a spherical 1-distance set $\leq h + 1$). When $N > h + 1$, all $N$ points collapse into a single point and $\beta = 1$, which is Case 1. When $N = h + 1$, these points either form a regular simplex or collapse into a single point, which is Case 2. When $N < h + 1$, these points form a regular/non-regular equidistant simplex or collapse into a single point, which is Case 3.

Next, we will show why when $K < h + 1$, $-\frac{1}{K-1} \leq \beta \leq 1$ (case 3) and when $Z = (z_1, \ldots, z_K) \in (\mathbb{S}^{h-1})^K$ forms a regular simplex, $\beta = -\frac{1}{K-1}$ (Case 2). Given that

$$\left\| \sum_{k \in [K]} z_k \right\|^2 = \left\langle \sum_{k \in [K]} z_k, \sum_{k \in [K]} z_k \right\rangle$$

$$= \sum_{k \in [K]} \langle z_k, z_k \rangle + \sum_{\substack{n \in [K] \\ m \in [K] \setminus \{i\}}} \langle z_n, z_m \rangle \tag{21}$$

$$= K + K(K-1)\beta$$

$$\geq 0,$$

this shows $-\frac{1}{K-1} \leq \beta$. Since $\beta$ is the dot product of two unit vectors, $\beta \leq 1$. Then we have:

$$-\frac{1}{N-1} \leq \beta \leq 1. \tag{22}$$

When $Z = (z_1, \ldots, z_K) \in (\mathbb{S}^{h-1})^K$ forms a regular simplex, we have $\sum_{k \in [K]} z_k = 0$. Then $K + K(K-1)\beta = 0$ and $\beta = -\frac{1}{K-1}$.

Now we prove when $\beta = -\frac{1}{K-1}$, $Z = (z_1, \ldots, z_K)$ forms a regular simplex. Recall that $\forall i, j \in [K]$ and $i \neq j$, we have $\|z_i\| = 1$, and $\langle z_i, z_j \rangle = \beta$. When $\beta = -\frac{1}{K-1}$, Eq. (22) shows $\sum_{k \in [K]} z_k = 0$. Then $Z$ forms a regular simplex. $\qquad \square$

**Lemma 2.** *Let $Z$ be an $N$-point configuration (assuming that all $z$s are normalized), $Z = (z_1, \ldots, z_N) \in (\mathbb{S}^{h-1})^N$, with labels $Y = (y_1, \ldots, y_N) \in ([K])^N$. Let $B = [N]$, $B_k = \{i : i \in B, y_i = k\}$. $\forall k \in [K]$, $\sum_{i \in B_k} \mathcal{L}_{\mathrm{SC}}^{k,i}$ is bounded below by:*

$$\sum_{i \in B_k} \mathcal{L}_{\mathrm{SC}}^{k,i} \geq N_k \log \left( (N_k - 1) + e^{-1} \sum_{\substack{k' \in [K] \\ k' \neq k}} N_{k'} \exp\left( \langle \bar{z}_k, \bar{z}_{k'} \rangle \right) \right), \tag{23}$$

*where $\bar{z}_k = \frac{1}{N_k} \sum_{i \in B_k} z_i$, and equality is attained if and only if the following condition holds:*

*(B1)* $\forall i \in B_k$, $z_i = \bar{z}_k$.

*Proof.* According to Eq. (17):

$$
\begin{aligned}
\mathcal{L}_{\mathrm{SC}}^{k,i} &= \log \left( \frac{\sum_{l \in B \setminus \{i\}} \exp\left( \langle z_i, z_l \rangle \right)}{\exp\left( \frac{1}{N_k - 1} \sum_{j \in B_k \setminus \{i\}} \langle z_i, z_j \rangle \right)} \right) \\
&= \log \left( \frac{\sum_{l \in B_k \setminus \{i\}} \exp\left( \langle z_i, z_l \rangle \right) + \sum_{\substack{k' \in [K] \\ k' \neq k}} \sum_{m \in B_{k'}} \exp\left( \langle z_i, z_m \rangle \right)}{\exp\left( \frac{1}{N_k - 1} \sum_{j \in B_k \setminus \{i\}} \langle z_i, z_j \rangle \right)} \right).
\end{aligned}
\tag{24}
$$

There are three terms in Eq. (24). Let's check their lower bounds one by one. Applying Jensen's inequity, the first term can be bounded below:

$$\sum_{l \in B_k \setminus \{i\}} \exp\left( \langle z_i, z_l \rangle \right) \geq (N_k - 1) \exp\left( \frac{1}{(N_k - 1)} \sum_{l \in B_k \setminus \{i\}} \langle z_i, z_l \rangle \right), \tag{25}$$

where equality is attained if and only if all of the following conditions hold:

*(C1)* $\forall k \in [K]$ and $\forall i \in B_k$, $\exists \alpha(k, i)$ such that $\forall j \in B_k \setminus \{i\}$, all inner products $\langle z_i, z_j \rangle = \alpha(k, i)$ are equal.

Let $\bar{z}_k = \frac{1}{N_k} \sum_{i \in B_k} z_i$. Similarly, the second term can be bounded below:

$$
\begin{aligned}
\sum_{m \in B_{k'}} \exp\left( \langle z_i, z_m \rangle \right) &\geq N_{k'} \exp\left( \frac{1}{N_{k'}} \sum_{m \in B_{k'}} \langle z_i, z_m \rangle \right) = N_{k'} \exp\left( \left\langle z_i, \frac{1}{N_{k'}} \sum_{m \in B_{k'}} z_m \right\rangle \right), \\
&= N_{k'} \exp\left( \langle z_i, \bar{z}_{k'} \rangle \right)
\end{aligned}
\tag{26}
$$

where equality is attained if and only if all of the following conditions hold:

*(C2)* $\forall k \in [K]$ and $\forall i \in B_k$, $\exists \alpha(k, i, k')$ such that $k' \in [K] \setminus \{k\}$ and $m \in B_{k'}$, all inner products $\langle z_i, z_m \rangle = \alpha'(k, i, k')$ are equal. And $\alpha'(k, i, k') = \langle z_i, \bar{z}_{k'} \rangle$.

Using Cauchy-Schwarz inequality, the third term can be bounded below:

$$\frac{1}{\exp\left( \frac{1}{N_k - 1} \sum_{j \in B_k \setminus \{i\}} \langle z_i, z_j \rangle \right)} \geq \frac{1}{\exp\left( \frac{1}{N_k - 1} \sum_{j \in B_k \setminus \{i\}} \|z_i\| \|z_j\| \right)} = e^{-1}, \tag{27}$$

where equality is attained if and only if the following condition holds:

(C3) $\forall k \in [K]$ and $\forall i, j \in B_k$, $z_i = z_j = \bar{z}_k$.

It is obvious to see that when condition (C3) holds, all samples from the same class collapse into their class center (denoted by $\bar{z}_k$). In this case, and thus condition (C1) and (C2) hold as well, where $\alpha(k, i) = 1$ and $\alpha'(k, i, k') = \langle z_k, \bar{z}_{k'} \rangle$. So, (C3) is a sufficient condition for (C1) and (C2). Now we have:

$$
\sum_{i \in B_k} \mathcal{L}_{\text{SC}}^{k,i} = \sum_{i \in B_k} \log \left( \frac{\sum_{k \in B_k \setminus \{i\}} \exp\left( \langle z_i, z_l \rangle \right) + \sum_{\substack{k' \in [K] \\ k' \neq k}} \sum_{l \in B_{k'}} \exp\left( \langle z_i, z_l \rangle \right)}{\exp\left( \frac{1}{N_k - 1} \sum_{j \in B_k \setminus \{i\}} \langle z_i, z_j \rangle \right)} \right)
$$

$$
\overset{Eq. (25)}{\geq} \sum_{i \in B_k} \log \left( \frac{(N_k - 1) \exp\left( \frac{1}{(N_k - 1)} \sum_{l \in B_k \setminus \{i\}} \langle z_i, z_l \rangle \right) + \sum_{\substack{k' \in [K] \\ k' \neq k}} \sum_{l \in B_{k'}} \exp\left( \langle z_i, z_l \rangle \right)}{\exp\left( \frac{1}{N_k - 1} \sum_{j \in B_k \setminus \{i\}} \langle z_i, z_j \rangle \right)} \right)
$$

$$
= \sum_{i \in B_k} \log \left( (N_k - 1) + \frac{\sum_{\substack{k' \in [K] \\ k' \neq k}} \sum_{l \in B_{k'}} \exp\left( \langle z_i, z_l \rangle \right)}{\exp\left( \frac{1}{N_k - 1} \sum_{j \in B_k \setminus \{i\}} \langle z_i, z_j \rangle \right)} \right)
$$

$$
\overset{Eq. (26)}{\geq} \sum_{i \in B_k} \log \left( (N_k - 1) + \frac{\sum_{\substack{k' \in [K] \\ k' \neq k}} N_{k'} \exp\left( \langle z_i, \bar{z}_{k'} \rangle \right)}{\exp\left( \frac{1}{N_k - 1} \sum_{j \in B_k \setminus \{i\}} \langle z_i, z_j \rangle \right)} \right)
$$

$$
\overset{Eq. (27)}{\geq} \sum_{i \in B_k} \log \left( (N_k - 1) + e^{-1} \sum_{\substack{k' \in [K] \\ k' \neq k}} N_{k'} \exp\left( \langle \bar{z}_k, \bar{z}_{k'} \rangle \right) \right)
$$

$$
= N_k \log \left( (N_k - 1) + e^{-1} \sum_{\substack{k' \in [K] \\ k' \neq k}} N_{k'} \exp\left( \langle \bar{z}_k, \bar{z}_{k'} \rangle \right) \right),
$$

$$
\tag{28}
$$

where equality is attained if and only if the following condition holds:

(B1) $\forall i \in B_k$, $z_i = \bar{z}_k$.

Here, (B1) and (C3) express the same condition. $\qquad\square$

**Lemma 3.** *Let $\bar{Z}$ be an $K$ point configuration (assuming all $\bar{z}s$ are normalized), $\bar{Z} = (\bar{z}_1, \ldots, \bar{z}_K) \in (\mathbb{S}^{h-1})^K$, and $K \le h + 1$, it holds that:*

$$\sum_{k=1}^{K} a \log \left( (a-1) + b \left( \sum_{\substack{k' \in [K] \\ k' \neq k}} \exp\left(\langle \bar{z}_k, \bar{z}_{k'} \rangle\right) + c \right) \right) \ge K a \log\left((a-1) + b\left((K-1)\exp(\beta) + c\right)\right), \tag{29}$$

*where $a > 1$, $b, c > 0$, and equality is attained if and only if all of the following conditions hold:*

(B2) $\forall k \in [K]$ *and* $k' \in [K]\backslash\{k\}$, $\langle \bar{z}_k, \bar{z}_{k'} \rangle = \beta$.

(B3) *There exists a configuration of $\bar{Z} = (\bar{z}_1, \ldots, \bar{z}_K)$ such that (B2) holds.*

    (Case 1) $K = h + 1$: $\beta = -\frac{1}{N-1}$ *or* $\beta = 1$

    (Case 2) $K < h + 1$: $-\frac{1}{N-1} \le \beta \le 1$

*Proof.* Since $f(x) = \exp(x)$ is a convex function, applying Jensen's inequality, we have

$$\begin{aligned}
\sum_{\substack{k' \in [K] \\ k' \neq k}} \exp\left(\langle \bar{z}_k, \bar{z}_{k'} \rangle\right) &\ge (K-1)\exp\left( \frac{1}{K-1} \sum_{\substack{k' \in [K] \\ k' \neq k}} \langle \bar{z}_k, \bar{z}_{k'} \rangle \right) \\
&= (K-1)\exp\left( \frac{1}{K-1} \sum_{\substack{k' \in [K] \\ k' \neq k}} \beta_k \right) \\
&= (K-1)\exp\left(\beta_k\right),
\end{aligned} \tag{30}$$

where equality is attained if and only if all of the following conditions hold:

(C4) $\forall k \in [K]$ *and* $k' \in [K]\backslash\{k\}$, $\langle \bar{z}_k, \bar{z}_{k'} \rangle = \beta_k$.

(C5) There exists a configuration of $\bar{Z} = (\bar{z}_1, \ldots, \bar{z}_K)$ such that (C4) holds.

When $a > 1, b, c > 0$, $f(x) = \log\left((a-1) + b\left(\exp(x) + c\right)\right)$ is also a convex function. By Jensen's inequality, we have

$$\begin{aligned}
\sum_{k=1}^{K} a \log\left((a-1) + b\left(\exp(\beta_k) + c\right)\right) &\ge K a \log\left( (a-1) + b\left( \exp\left( \frac{1}{K} \sum_{k=1}^{K} \beta_k \right) + c \right) \right) \\
&= K a \log\left( (a-1) + b\left( \exp\left( \frac{1}{K} \sum_{k=1}^{K} \beta \right) + c \right) \right) \\
&= K a \log\left((a-1) + b\left(\exp(\beta) + c\right)\right),
\end{aligned} \tag{31}$$

where equality is attained if and only if all of the following conditions hold:

(C6) $\forall k \in [K]$ *and* $k' \in [K]\backslash\{k\}$, $\beta_k = \beta_{k'} = \beta$.

(C7) There exists a configuration of $\bar{Z} = (\bar{z}_1, \ldots, \bar{z}_K)$ such that (C6) holds.

Note that when (C6) and (C7) hold, (C4) and (C5) hold too. And according to Lemma 1, when $K \leq h + 1$, Case 2 and Case 3 in Lemma 1 satisfy (C7). And hence

$$\sum_{k=1}^{K} a \log \left( (a-1) + b \left( \sum_{\substack{k' \in [K] \\ k' \neq k}} \exp \left( \langle \bar{z}_k, \bar{z}_{k'} \rangle \right) + c \right) \right) \geq K a \log \left( (a-1) + b \left( (K-1) \exp \left( \beta \right) + c \right) \right),$$

(32)

where equality is attained if and only if all of the following conditions hold:

(B2) $\forall k \in [K]$ and $k' \in [K] \backslash \{k\}$, $\langle \bar{z}_k, \bar{z}_{k'} \rangle = \beta$.

(B3) There exists a configuration of $\bar{Z} = (\bar{z}_1, \ldots, \bar{z}_K)$ such that (B2) holds.

(Case 1) $K = h + 1$: $\beta = -\frac{1}{N-1}$ or $\beta = 1$

(Case 2) $K < h + 1$: $-\frac{1}{N-1} \leq \beta \leq 1$

$\square$

## B.2 Proof of Theorem 2

In this section, we provide proofs for Theorem 2 proposed in Sec. 3.2. For convenience in reading, let us recall some related notions and definitions.

- $h, N, K \in \mathbb{N}$
- $\mathcal{Z} = \mathbb{R}^h$
- $\mathbb{S}^{h-1} = \left\{ z \in \mathbb{R}^h : \|z\| = 1 \right\}$
- $\mathcal{Y} = \{1, \ldots, K\} = [K]$
- $B = \{1, \ldots, N\} = [N]$
- $B_k = \{i : i \in B, y_i = k\}$
- $N_k = |B_k|$

**Definition 1** (Supervised contrastive loss) Let $Z$ be an $N$-point configuration (assuming that all $zs$ are normalized), $Z = (z_1, \ldots, z_N) \in (\mathbb{S}^{h-1})^N$, with labels $Y = (y_1, \ldots, y_N) \in ([K])^N$, and $K \leq h+1$. Let $B = [N]$, $B_k = \{i : i \in B, y_i = k\}$ and $N_k = |B_k|$. Supervised contrastive loss $\mathcal{L}_{SC}(\cdot; Y) : (\mathbb{S}^{h-1})^N \to \mathbb{R}$ is defined as:

$$\mathcal{L}_{SC} = \sum_{k=1}^{K} \sum_{i \in B_k} \mathcal{L}_{SC}^{k,i}, \text{ where } \mathcal{L}_{SC}^i = -\frac{\mathbb{1}_{\{N_k > 1\}}}{N_k - 1} \sum_{j \in B_k \setminus \{i\}} \log \left( \frac{\exp\left(\langle z_i, z_j \rangle\right)}{\sum_{l \in B \setminus \{i\}} \exp\left(\langle z_i, z_l \rangle\right)} \right).$$

**Theorem 2** Let $Z$ be an $N$-point configuration (assuming that all $zs$ are normalized), $Z = (z_1, \ldots, z_N) \in (\mathbb{S}^{h-1})^N$, with labels $Y = (y_1, \ldots, y_N) \in ([K])^N$, and $3 \leq K \leq h+1$. If $\forall k \in \{2, \ldots, K\}, N_k = a_2 \geq 4$, and $\exists \rho > 0$ such that $N_1 = a_1 = \rho a_2 > 1$, it holds that:

$$\mathcal{L}_{SC} \geq f(\cos(\theta_1), \cos(\theta_2)),$$

where $f(\cdot) : \mathbb{R} \times \mathbb{R} \to \mathbb{R}$ is defined as:

$$
\begin{aligned}
f(x_1, x_2) = {} & \rho a_2 \log\left( (\rho a_2 - 1) + e^{-1} (K-1) a_2 \exp(x_1) \right) \\
& + (K-1) a_2 \log\left( (a_2 - 1) + e^{-1} \left( (K-2) a_2 \exp(x_2) + \rho a_2 \exp(x_1) \right) \right),
\end{aligned}
$$

and equality is attained if and only if there exists a configuration of $\bar{Z} = (\bar{z}_1, \ldots, \bar{z}_K) \in (\mathbb{S}^{h-1})^K$ such that:

(A3) $i \in B_k, z_i = \bar{z}_k$.

(A4) $(\bar{z}_2, \ldots, \bar{z}_K)$ form an equidistant simplex whose vertex–center–vertex angle equals $\theta_2$.

(A5) $\forall k \in \{2, \ldots, K\}, \langle \bar{z}_1, \bar{z}_k \rangle = \cos(\theta_1)$ and $\cos(\theta_2) = \frac{(K-1)\cos^2(\theta_1) - 1}{K-2}$.

The numerical relationship between $\rho$ and $\theta_1$ can be summarized as:

(Case 1) $\rho < 1$: $\theta_1 \in \left( \cos^{-1}(-\frac{1}{K-1}), 0 \right)$

(Case 2) $\rho = 1$: $\theta_1 = \cos^{-1}(-\frac{1}{K-1})$.

(Case 3) $1 < \rho < R(K, a_2)$: $\theta_1 \in \left( -\pi, \cos^{-1}(-\frac{1}{K-1}) \right)$

(Case 4) $\rho \geq R(K, a_2)$: $\theta_1 = -\pi$.

Let $b_1 = (K-1)(1 + e^{-2} - 2e^2)a_2 - 2$, $b_2 = 8(1 + e^{-2})(K-1)a_2((K-1)a_2 - e^2)$, then $R(K, a_2)$ be defined as:

$$R(K, a_2) = \frac{-b_1 + \sqrt{b_1^2 + b_2}}{2(1 + e^{-2})a_2}.$$

### B.2.1 STEPS OF PROOF

Following Eq. (17), Eq. (18) and Lemma 2 in Appendix B.1.1, we have

$$\mathcal{L}_{\text{SC}} \overset{Lemma\ 2}{\geq} \sum_{k=1}^{K} N_k \log \left( (N_k - 1) + e^{-1} \sum_{\substack{k' \in [K] \\ k' \neq k}} N_{k'} \exp \left( \langle \bar{z}_k, \bar{z}_{k'} \rangle \right) \right). \tag{33}$$

where equality is attained if and only if there exists a configuration of $\bar{Z} = (\bar{z}_1, \ldots, \bar{z}_K) \in (\mathbb{S}^{h-1})^K$ such that:

(A3)  $i \in B_k$, $z_i = \bar{z}_k$.

When $3 \leq K \leq h + 1$, $\forall k \in \{2, \ldots, K\}$, $N_k = a_2 \geq 4$, and $\exists \rho > 0$ such that $N_1 = a_1 = \rho a_2 > 1$, following Lemma 5, we have:

$$\mathcal{L}_{\text{SC}} \geq \sum_{k=1}^{K} N_k \log \left( (N_k - 1) + e^{-1} \sum_{\substack{k' \in [K] \\ k' \neq k}} N_{k'} \exp \left( \langle \bar{z}_k, \bar{z}_{k'} \rangle \right) \right)$$

$$\overset{Lemma\ 5}{\geq} f(\cos(\theta_1), \cos(\theta_2)), \tag{34}$$

where $f(\cdot) : \mathbb{R} \times \mathbb{R} \to \mathbb{R}$ is defined as:

$$\begin{aligned}
f(x_1, x_2) &= \rho a_2 \log \left( (\rho a_2 - 1) + e^{-1} (K - 1) a_2 \exp(x_1) \right) \\
&\quad + (K - 1) a_2 \log \left( (a_2 - 1) + e^{-1} ((K - 2) a_2 \exp(x_2) + \rho a_2 \exp(x_1)) \right),
\end{aligned} \tag{35}$$

and equality is attained if and only if all of the following conditions hold:

(A4)  $(\bar{z}_2, \ldots, \bar{z}_K)$ form an equidistant simplex whose vertex–center–vertex angle equals $\theta_2$.

(A5)  $\forall k \in \{2, \ldots, K\}$, $\langle \bar{z}_1, \bar{z}_k \rangle = \cos(\theta_1)$ and $\cos(\theta_2) = \frac{(K-1)\cos^2(\theta_1) - 1}{K - 2}$.

The numerical relationship between $\rho$ and $\theta_1$ can be summarized as:

(Case 1)  $\rho < 1$: $\theta_1 \in \left( \cos^{-1}(-\frac{1}{K-1}), 0 \right)$

(Case 2)  $\rho = 1$: $\theta_1 = \cos^{-1}(-\frac{1}{K-1})$.

(Case 3)  $1 < \rho < R(K, a_2)$: $\theta_1 \in \left( -\pi, \cos^{-1}(-\frac{1}{K-1}) \right)$

(Case 4)  $\rho \geq R(K, a_2)$: $\theta_1 = -\pi$.

Let $b_1 = (K-1)(1 + e^{-2} - 2e^2)a_2 - 2$, $b_2 = 8(1 + e^{-2})(K-1)a_2((K-1)a_2 - e^2)$, then $R(K, a_2)$ be defined as:

$$R(K, a_2) = \frac{-b_1 + \sqrt{b_1^2 + b_2}}{2(1 + e^{-2})a_2}.$$

### B.2.2 LEMMAS PART 2

In this section, we provide definitions and proofs of lemmas that are used for the proof of Theorem 2.

**Lemma 4.** *Let $\bar{Z}$ be an $K$ point configuration (assuming that all $\bar{z}$s are normalized), $\bar{Z} = (\bar{z}_1, \ldots, \bar{z}_K) \in (\mathbb{S}^{h-1})^K$, and $3 \leq K \leq h+1$. If $\forall k, k' \in \{2, \ldots, K\}$ and $k \neq k'$ such that $\langle \bar{z}_k, \bar{z}_{k'} \rangle = \beta_2$ and $\beta_1 = min_c\{c : \langle \bar{z}_1, \bar{z}_k \rangle = \langle \bar{z}_1, \bar{z}_{k'} \rangle = c\}$, it holds that:*

$$\beta_2 = \frac{(K-1)\beta_1^2 - 1}{K-2}, \text{ where } -1 \leq \beta_1 \leq 0 \text{ and } -\frac{1}{K-2} \leq \beta_2 < 1. \tag{36}$$

*Proof.* Without loss of generality, we assume that $(\bar{z}_2, \ldots, \bar{z}_K)$ forms an equidistant simplex in the southern hemisphere of $\mathbb{S}^{h-1}$ and then $\bar{z}_1$ is at the north pole. Let $l = \|\frac{1}{K-1}\sum_{k=2}^{K} z_k\|$, we have $l = |\beta_1|$, then

$$
\begin{aligned}
\|l\|^2 &= \left\|\frac{1}{K-1}\sum_{k=2}^{K} z_k\right\|^2 = \left\langle \frac{1}{K-1}\sum_{k=2}^{K} z_k, \frac{1}{K-1}\sum_{k=2}^{K} z_k \right\rangle \\
&= \frac{1}{(K-1)^2}\left( \sum_{k=2}^{K} z_k \langle z_k, z_k \rangle + \sum_{\substack{k,k'=2 \\ k \neq k'}}^{K} \langle z_k, z_{k'} \rangle \right) \\
&= \frac{1}{(K-1)^2}\left( (K-1) + (K-1)(K-2)\beta_2 \right) \\
&= |\beta_1|^2,
\end{aligned}
\tag{37}
$$

so we have

$$\beta_2 = \frac{(K-1)\beta_1^2 - 1}{K-2}. \tag{38}$$

According to Lemma 1, $-\frac{1}{K-2} \leq \beta_2 < 1$ and so $-1 \leq \beta_1 \leq 0$. $\qquad\square$

**Lemma 5.** *Let $\bar{Z}$ be an $K$ point configuration (assuming all $\bar{z}$s are normalized), $\bar{Z} = (\bar{z}_1, \ldots, \bar{z}_K) \in (\mathbb{S}^{h-1})^K$, and $3 \leq K \leq h+1$. Let $B = [N]$, $B_k = \{i : i \in B, y_i = k\}$ and $N_k = |B_k|$. Let $\mathcal{J}(\cdot) : (\mathbb{S}^{h-1})^K \to \mathbb{R}$ is defined as:*

$$\mathcal{J}(\bar{Z}) = \sum_{k=1}^{K} N_k \log \left( (N_k - 1) + e^{-1} \sum_{\substack{k' \in [K] \\ k' \neq k}} N_{k'} \exp\left(\langle \bar{z}_k, \bar{z}_{k'} \rangle\right) \right), \tag{39}$$

*If $\forall k \in \{2, \ldots, K\}, N_k = a_2 \geq 4$, and $\exists \rho > 0$ such that $N_1 = a_1 = \rho a_2 > 1$, it holds that:*

$$\mathcal{J}(\bar{Z}) \geq f(\beta_1), \tag{40}$$

*where $f(\cdot) : \mathbb{R} \to \mathbb{R}$ is defined as:*

$$
\begin{aligned}
f(x) = {} & \rho a_2 \log\left( (\rho a_2 - 1) + e^{-1}(K-1)a_2 \exp(x) \right) \\
& + (K-1)a_2 \log\left( (a_2 - 1) + e^{-1}\left( (K-2)a_2 \exp\left(\frac{(K-1)x^2 - 1}{K-2}\right) + \rho a_2 \exp(x) \right) \right),
\end{aligned}
\tag{41}
$$

*and equality is attained if and only if all of the following condition holds:*

*(B4)* $\forall k, k' \in \{2, \dots, K\}$ *and* $k \neq k'$, $\langle \bar{z}_1, \bar{z}_k \rangle = \beta_1$ *and* $\langle \bar{z}_k, \bar{z}_{k'} \rangle = \beta_2 = \frac{(K-1)\beta_1^2 - 1}{K-2}$.

*Let* $\beta_1 = \cos(\theta_1)$, *the numerical relationship between* $\rho$ *and* $\theta_1$ *can be summarized as:*

*(Case 1)* $\rho < 1$: $\theta_1 \in (-\frac{1}{K-1}, 0)$.

*(Case 2)* $\rho = 1$: $\theta_1 = -\frac{1}{K-1}$.

*(Case 3)* $1 < \rho < R(K, a_2)$: $\theta_1 \in (-1, -\frac{1}{K-1})$.

*(Case 4)* $\rho \geq R(K, a_2)$: $\theta_1 = -1$.

*Here* $b_1 = (K-1)(1+e^{-2}-2e^2)a_2 - 2$ *and* $b_2 = 8(1+e^{-2})(K-1)a_2((K-1)a_2 - e^2)$. $R(K, a_2)$ *is given by:*

$$R(K, a_2) = \frac{-b_1 + \sqrt{b_1^2 + b_2}}{2(1+e^{-2})a_2}.\tag{42}$$

*Proof.* When $N_1 = a_1, \forall k \in \{2, \dots, K\}, N_k = a_2, a_1 = \rho a_2$, then

$$
\begin{aligned}
\mathcal{J}(\bar{Z}) &= \sum_{k=1}^{K} N_k \log \left( (N_k - 1) + e^{-1} \sum_{\substack{k' \in [K] \\ k' \neq k}} N_{k'} \exp\left(\langle \bar{z}_k, \bar{z}_{k'} \rangle\right) \right) \\
&= a_1 \log \left( (a_1 - 1) + e^{-1} \sum_{k'=2}^{K} a_2 \exp\left(\langle \bar{z}_1, \bar{z}_{k'} \rangle\right) \right) \\
&\quad + \sum_{k=2}^{K} a_2 \log \left( (a_2 - 1) + e^{-1} \left( \sum_{\substack{k'=2 \\ k' \neq k}}^{K} a_2 \exp\left(\langle \bar{z}_k, \bar{z}_{k'} \rangle\right) + a_1 \exp\left(\langle \bar{z}_k, \bar{z}_1 \rangle\right) \right) \right).
\end{aligned}\tag{43}
$$

According to Eq. (30) in Lemma 3, the first term can be bounded low:

$$
\begin{aligned}
&a_1 \log \left( (a_1 - 1) + e^{-1} \sum_{k'=2}^{K} a_2 \exp\left(\langle \bar{z}_1, \bar{z}_{k'} \rangle\right) \right) \\
&\geq a_1 \log \left( (a_1 - 1) + e^{-1}(K-1)a_2 \exp\left(\beta_1\right) \right) \\
&= f_1(\beta_1).
\end{aligned}\tag{44}
$$

Similarly, the second term can be bounded low:

$$
\begin{aligned}
&\sum_{k=2}^{K} a_2 \log \left( (a_2 - 1) + e^{-1} \left( \sum_{\substack{k'=2 \\ k' \neq k}}^{K} a_2 \exp\left(\langle \bar{z}_k, \bar{z}_{k'} \rangle\right) + a_1 \exp\left(\langle \bar{z}_k, \bar{z}_1 \rangle\right) \right) \right) \\
&\geq (K-1)a_2 \log \left( (a_2 - 1) + e^{-1}\left( (K-2)a_2 \exp\left(\beta_2\right) + a_1 \exp\left(\beta_1\right) \right) \right) \\
&= f_2(\beta_1).
\end{aligned}\tag{45}
$$

Combining Eq. (44), Eq. (45) and Lemma 4, we have

$$\mathcal{J}(\bar{Z}) \geq \rho a_2 \log \left( (\rho a_2 - 1) + e^{-1} (K-1) a_2 \exp(\beta_1) \right)$$
$$+ (K-1)a_2 \log \left( (a_2 - 1) + e^{-1} \left( (K-2) a_2 \exp\left( \frac{(K-1)\beta_1^2 - 1}{K-2} \right) + \rho a_2 \exp(\beta_1) \right) \right)$$
$$= f_1(\beta_1) + f_2(\beta_1) = f(\beta_1),$$
(46)

where $-1 \leq \beta_1 \leq 0$ and equality is attained if and only if the following condition holds:

(C8) $\forall k, k' \in \{2, \ldots, K\}$ and $k \neq k'$, $\langle \bar{z}_1, \bar{z}_k \rangle = \beta_1$ and $\langle \bar{z}_k, \bar{z}_{k'} \rangle = \beta_2 = \frac{(K-1)\beta_1^2 - 1}{K-2}$.

To find the minimal value of $f(x)$ when $-1 \leq x \leq 0$, we need to find the critical value of $f'(x) = 0$ and the sign of $f'(x)$. Direct computation of these value is difficult but can be found with scientific computation software once we know all parameters in a specific case. For analytical purpose, we investigate the general form. Let $3 \leq K \leq h+1, \rho > 0, a_1 = \rho a_2 > 1, a_2 \geq 4$ and $-1 \leq x \leq 0$.

We first study key properties of $f(x)$.

(P1) We start by analyzing the derivatives of $f(x)$. The first and the second derivative of $f_1(x)$ are:

$$f_1'(x) = e^{-1}(K-1)a_2^2 \frac{\rho e^x}{(\rho a_2 - 1) + e^{-1}(K-1)a_2 e^x} > 0,$$
(47)

and

$$f_1''(x) = e^{-1}(K-1)a_2^2 \frac{(\rho a_2 - 1)\rho e^x}{((\rho a_2 - 1) + e^{-1}(K-1)a_2 e^x)^2} > 0.$$
(48)

Here $f_1'(x)$ and $f_1''(x)$ are strictly positive because every term of them is positive. The First derivative of $f_2(x)$ is:

$$f_2'(x) = e^{-1}(K-1)a_2^2 \frac{2(K-1)x \exp\left( \frac{(K-1)x^2 - 1}{K-2} \right) + \rho e^x}{(a_2 - 1) + e^{-1} \left( (K-2)a_2 \exp\left( \frac{(K-1)x^2 - 1}{K-2} \right) + \rho a_2 e^x \right)}.$$
(49)

The second derivative of $f_2(x)$ is difficult to calculate directly. We instead do it in another way. If we take $y(x) = \exp(\frac{(K-1)x^2 - 1}{K-2})$ as a variable, we have:

$$\frac{dy(x)}{dx} = \frac{2(K-1)x}{K-2} \exp(\frac{(K-1)x^2 - 1}{K-2}) < 0.$$
(50)

It holds because every term but $x$ (negative) in $\frac{dy(x)}{dx}$ is positive. And $f_2'(x)$ can be written as:

$$f_2'(x) = G(x, y) = e^{-1}(K-1)a_2^2 \frac{\rho e^x + 2(K-1)xy}{(a_2 - 1) + e^{-1}\rho a_2 e^x + e^{-1}(K-2)a_2 y}$$
$$= c_1 \frac{c_2 + c_3 y}{c_4 + c_5 y},$$
(51)

where $c_1 = e^{-1}(K-1)a_2^2 > 0$, $c_2 = \rho e^x$, $c_3 = 2(K-1)x$, $c_4 = (a_2 - 1) + e^{-1}\rho a_2 e^x$, $c_5 = e^{-1}(K-2)a_2$ and $-1 \leq x \leq 0$. Then the partial derivative of $G$ to $y$ is:

$$\frac{\partial G(x, y)}{\partial y} = \frac{c_1}{(c_4 + c_5 y)^2} (c_3 c_4 - c_2 c_5)$$
$$= \frac{c_1}{(c_4 + c_5 y)^2} \left( (2(K-1)x - (K-2)) e^{-1}\rho a_2 e^x + (a_2 - 1)2(K-1)x \right)$$
$$< 0.$$
(52)

Here $\frac{\partial G}{\partial y}$ is strictly negative because $(2(K-1)x - (K-2))$ and $x$ are negative while all other terms are positive. Similarly, $f_2'(x)$ can be written as:

$$
\begin{aligned}
f_2'(x) = G(x, y) &= e^{-1}(K-1)a_2^2 \frac{2(K-1)yx + \rho e^x}{(a_2 - 1) + e^{-1}(K-2)a_2 y + e^{-1}\rho a_2 e^x} \\
&= c_1 \frac{c_6 x + c_7 e^x}{c_8 + c_9 e^x},
\end{aligned}
\tag{53}
$$

where $c_1 = e^{-1}(K-1)a_2^2$, $c_6 = 2(K-1)y$, $c_7 = \rho$, $c_8 = (a_2 - 1) + e^{-1}(K-2)a_2 y$, $c_9 = e^{-1}\rho a_2$ and $-1 \leq x \leq 0$. Here $c_1, c_6, c_7, c_8, c_9 > 0$. Then the partial derivative of $G$ to $x$ is:

$$
\frac{\partial G(x, y)}{\partial x} = \frac{c_1}{(c_8 + c_9 e^x)^2} \left( (1 - x)c_6 c_9 e^x + c_7 c_8 e^x + c_6 c_8 \right) > 0.
\tag{54}
$$

Here $\frac{\partial G}{\partial x}$ is strictly positive because every term of it is positive. Combining Eq. (50), Eq. (52) and Eq. (54), we have:

$$
f_2''(x) = \frac{\partial G(x, y)}{\partial x} + \frac{\partial G(x, y)}{\partial y} \cdot \frac{dy(x)}{x} > 0.
\tag{55}
$$

Thus, according to Eq. (48) and Eq. (55), the second derivative of $f(x)$ is:

$$
f''(x) = f_1''(x) + f_2''(x) > 0.
\tag{56}
$$

This reveals that $f(x)$ is a convex function.

**(P2)** Next, we analyze how $\rho$ affects $f'(x)$. If we view $\rho$ as a variable instead of a constant, we have

$$
\begin{aligned}
f_1'(x) = H_1(x, \rho) &= e^{-1}(K-1)a_2^2 e^x \frac{\rho}{a_2 \rho + e^{-1}(K-1)a_2 e^x - 1} \\
&= c_1 \frac{\rho}{a_2 \rho + c_2},
\end{aligned}
\tag{57}
$$

where $c_1 = e^{-1}(K-1)a_2^2 e^x > 0$, $c_2 = e^{-1}(K-1)a_2 e^x - 1$. Then the partial derivative of $H_1$ to $\rho$ is given by:

$$
\begin{aligned}
\frac{\partial H_1(x, \rho)}{\partial \rho} &= c_1 \frac{c_2}{(a_2 \rho + c_2)^2} = c_1 \frac{e^{-1}(K-1)a_2 e^x - 1}{(a_2 \rho + c_2)^2} \\
&> c_1 \frac{e^{-1}(3-1)a_2 e^{-1} - 1}{(a_2 \rho + c_2)^2} = c_1 \frac{2e^{-2}a_2 - 1}{(a_2 \rho + c_2)^2} \\
&> 0.
\end{aligned}
\tag{58}
$$

When $K \geq 3$ and $-1 \leq x \leq 0$, $e^{-1}(K-1)a_2 e^x > 2e^{-2}a_2$. So, as long as $a_2 \geq 4 > \frac{e^2}{2}$ holds, $c_2 > 0$ holds. Similarly, we also have:

$$
\begin{aligned}
f_2'(x) = H_2(x, \rho) &= e^{-1}(K-1)a_2^2 e^x \frac{\rho + 2(K-1)x e^{-x} \exp\left( \frac{(K-1)x^2 - 1}{K-2} \right)}{e^{-1}a_2 e^x \rho + (a_2 - 1) + e^{-1}(K-2)a_2 \exp\left( \frac{(K-1)x^2 - 1}{K-2} \right)} \\
&= c_1 \frac{\rho + c_3}{c_4 \rho + c_5},
\end{aligned}
\tag{59}
$$

where $c_1 = e^{-1}(K-1)a_2^2 e^x > 0$, $c_3 = 2(K-1)xe^{-x}\exp\left(\frac{(K-1)x^2-1}{K-2}\right)$, $c_4 = e^{-1}a_2 e^x$ and $c_5 = (a_2-1) + e^{-1}(K-2)a_2\exp\left(\frac{(K-1)x^2-1}{K-2}\right)$. Then the partial derivative of $H_2$ to $\rho$ is:

$$\frac{\partial H_2(x,\rho)}{\partial \rho} = c_1 \frac{c_5 - c_3 c_4}{(c_4\rho + c_5)^2} = c_1 \frac{(a_2-1) + e^{-1}a_2\left(-Kx\right)\exp\left(\frac{(K-1)x^2-1}{K-2}\right)}{(c_4\rho + c_5)^2} \tag{60}$$
$$> 0.$$

It holds because every term in $\frac{\partial H_2}{\partial \rho}$ is positive. Combining Eq. (57) to Eq. (60), we have

$$f'(x) = f_1'(x) + f_2'(x)$$
$$= H_1(x,\rho) + H_2(x,\rho) = H(x,\rho), \tag{61}$$

and

$$\frac{\partial H(x,\rho)}{\partial \rho} = \frac{\partial H_1(x,\rho)}{\partial \rho} + \frac{\partial H_2(x,\rho)}{\partial \rho} > 0. \tag{62}$$

So $f'(x) = H(x,\rho)$ is an increasing function with respect to $\rho$.

With the above 2 key properties of $f(x)$ in hand, let us check some important values.

**(V1).** When $x = 0$, we have:

$$f'(0) = \frac{e^{-1}(K-1)\rho a_2^2}{(\rho a_2 - 1) + e^{-1}(K-1)a_2} + \frac{e^{-1}(K-1)\rho a_2^2}{(a_2-1) + e^{-1}\left((K-2)a_2\exp\left(-\frac{1}{K-2}\right) + \rho a_2\right)} \tag{63}$$
$$> 0.$$

It holds because every term in $f'(0)$ is positive. This case shows that, when samples from $K-1$ equal-sized classes are well trained, they form a $K-2$ regular simplex ($\beta_1 = 0$ and $\beta_2 = -\frac{1}{K-2}$). Once samples from the $K^{\text{th}}$ class come, the original $K-2$ simplex starts to shrink as the loss goes down when $\beta_1$ decreases and $\beta_2$ increases.

**(V2).** When $x = -\frac{1}{K-1}$, we have:

$$f'(-\frac{1}{K-1}) = H(-\frac{1}{K-1},\rho)$$
$$= \frac{e^{-1}(K-1)a_2^2 e^{-\frac{1}{K-1}}\rho}{(\rho a_2 - 1) + e^{-1}(K-1)a_2 e^{-\frac{1}{K-1}}} + \frac{e^{-1}(K-1)a_2^2 e^{-\frac{1}{K-1}}(\rho-2)}{(a_2-1) + e^{-1}a_2 e^{-\frac{1}{K-1}}((K-2+\rho)}, \tag{64}$$

and

$$H(-\frac{1}{K-1},1) = 0. \tag{65}$$

According to Eq. (61) and Eq. (62), $H(-\frac{1}{K-1},\rho)$ is an increasing function with respect to $\rho$. Recalling that $f(x)$ is a convex function, with Eq. (64) and Eq. (65), we can conclude that:

(C9) When $\rho < 1$: $f'(-\frac{1}{K-1}) < H(-\frac{1}{K-1},1) = 0$. Since $f'(0) > 0$, according to the intermediate value theorem, there exists a critical point $\hat{x} \in (-\frac{1}{K-1},0)$, such that $f'(\hat{x}) = 0$, and $f(x)$ attains its minimal value at $x = \hat{x}$. If $\rho$ increases, $f'(\hat{x})$ increases too. It leads to $f'(\hat{x}) > 0$, then there comes a new critical point $\tilde{x} \in (-\frac{1}{K-1},\hat{x})$ where $f'(\tilde{x}) = 0$.

(C10) When $\rho = 1$: $f'(-\frac{1}{K-1}) = H(-\frac{1}{K-1}, 1) = 0$. So $\hat{x} = -\frac{1}{K-1}$ is the critical point and $f(x)$ attains its minimal value at $x = -\frac{1}{K-1}$.

(C11) When $\rho > 1$: $f'(-\frac{1}{K-1}) > H(-\frac{1}{K-1}, 1) = 0$. And $\forall x \in [-\frac{1}{K-1}, 0]$, $f'(x) > 0$ and $f(x) \geq f(-\frac{1}{K-1})$.

**(V3).** When $x = -1$, from (C7) and (C8) we know that $f'(-1) < 0$ if $\rho \leq 1$. Now let's only consider the case when $\rho > 1$.

$$
\begin{aligned}
f'(-1) &= \frac{e^{-2}(K-1)a_2^2\rho}{(\rho a_2 - 1) + e^{-2}(K-1)a_2} + (K-1)a_2^2 \frac{-2(K-1) + e^{-2}\rho}{(a_2 - 1) + (K-2)a_2 + e^{-2}\rho a_2} \\
&= e^{-2}(K-1)a_2^2 \left( \frac{\rho}{a_2\rho + e^{-2}(K-1)a_2 - 1} + \frac{\rho - 2(K-1)e^2}{e^{-2}a_2\rho + (K-1)a_2 - 1} \right) \\
&= e^{-2}(K-1)a_2^2 \left( \frac{\rho}{a_2\rho + c_1} + \frac{\rho + c_2}{e^{-2}a_2\rho + c_3} \right) \\
&= \frac{e^{-2}(K-1)a_2^2}{(a_2\rho + c_1)(e^{-2}a_2\rho + c_3)} \left( (1 + e^{-2})a_2\rho^2 + (c_1 + c_3 + a_2 c_2)\rho + c_1 c_2 \right) \\
&= \frac{(K-1)e^{-2}a_2^2}{(a_2\rho + c_1)(e^{-2}a_2\rho + c_3)} \cdot L(\rho),
\end{aligned}
\tag{66}
$$

where $c_1 = e^{-2}(K-1)a_2 - 1$, $c_2 = -2(K-1)e^2$, $c_3 = (K-1)a_2 - 1$ and:

$$
\begin{aligned}
L(\rho) &= (1 + e^{-2})a_2\rho^2 + (c_1 + c_3 + a_2 c_2)\rho + c_1 c_2 \\
&= (1 + e^{-2})a_2\rho^2 + ((K-1)(1 + e^{-2} - 2e^2)a_2 - 2)\rho - 2(K-1)((K-1)a_2 - e^2).
\end{aligned}
\tag{67}
$$

When $K \geq 3$, as long as $a_2 \geq 4 > \frac{e^2}{2}$ holds, $c_1 > 2e^{-2}a_2 - 1 > 0$. Also $c_3 > 0$, so we have $\frac{(K-1)e^{-2}a_2^2}{(a_2\rho + c_1)(e^{-2}a_2\rho + c_3)} > 0$, then $f'(-1) \geq 0 \Leftrightarrow L(\rho) \geq 0$. To solve this inequality, let us first take a look at the value:

$$
\begin{aligned}
M &= (c_1 + c_3 + a_2 c_2)^2 - 4(1 + e^{-2})a_2 c_1 c_2 \\
&> -4(1 + e^{-2})a_2 c_1 c_2 \\
&= 8(1 + e^{-2})a_2(K-1)e^2 c_1 \\
&> c_1 > 0.
\end{aligned}
\tag{68}
$$

Let $b_1 = c_1 + c_3 + a_2 c_2 = (K-1)(1 + e^{-2} - 2e^2)a_2 - 2 < 0$, $b_2 = -4(1 + e^{-2})a_2 c_1 c_2 = 8(1 + e^{-2})(K-1)a_2((K-1)a_2 - e^2) > 0$ and $M = b_1^2 + b_2 > 0$. Then the solution for $L(\rho) > 0$ and also $f'(-1) > 0$ is:

$$
\rho \leq \frac{-b_1 - \sqrt{b_1^2 + b_2}}{2(1 + e^{-2})a_2} \text{ or } \rho \geq \frac{-b_1 + \sqrt{b_1^2 + b_2}}{2(1 + e^{-2})a_2}.
\tag{69}
$$

Since $b_2 > 0$, then $\sqrt{b_1^2 + b_2} > -b_1$ and so $-b_1 - \sqrt{b_1^2 + b_2} < 0$. As we only consider the case where $\rho > 1$. We retain the right-hand part of Eq. (69).

Combined with (C11), now we can conclude that: when $K \geq 3$ and $a_2 \geq 4$, let

$$
R(K, a_2) = \frac{-b_1 + \sqrt{b_1^2 + b_2}}{2(1 + e^{-2})a_2},
\tag{70}
$$

where $b_1 = (K-1)(1 + e^{-2} - 2e^2)a_2 - 2 < 0$ and $b_2 = 8(1 + e^{-2})(K-1)a_2((K-1)a_2 - e^2) > 0$

(C12) When $1 < \rho < R(K, a_2)$: $f'(-1) < 0$. Since $\forall x \in [-\frac{1}{K-1}, 0]$, $f'(x) > 0$, according to the intermediate value theorem, there exists a critical point $\hat{x} \in (-1, -\frac{1}{K-1})$, such that $f'(\hat{x}) = 0$ and $f(x)$ attains its minimal value when $x = \hat{x}$. If $\rho$ increases, $f'(\hat{x})$ increases too. It leads to $f'(\hat{x}) > 0$, then there comes a new critical point $\tilde{x} \in (-1, \hat{x})$ where $f'(\tilde{x}) = 0$.

(C13) When $\rho \geq R(K, a_2)$: $f'(-1) \geq 0$. Then $\forall x \in [-1, 0]$, $f'(x) \geq 0$. $f(x)$ attains its minimal value when $x = -1$

Combining (C8) to (C13), we conclude that: $\mathcal{J}(\bar{Z})$ reach its minimal if and only if the following condition holds:

(B4) $\forall k, k' \in \{2, \ldots, K\}$ and $k \neq k'$, $\langle \bar{z}_1, \bar{z}_k \rangle = \beta_1$ and $\langle \bar{z}_k, \bar{z}_{k'} \rangle = \beta_2 = \frac{(K-1)\beta_1^2 - 1}{K-2}$.

Let $\beta_1 = \cos(\theta_1)$, the numerical relationship between $\rho$ and $\theta_1$ can be summarized as:

(Case 1) $\rho < 1$: $\theta_1 \in (-\frac{1}{K-1}, 0)$.

(Case 2) $\rho = 1$: $\theta_1 = -\frac{1}{K-1}$.

(Case 3) $1 < \rho < R(K, a_2)$: $\theta_1 \in (-1, -\frac{1}{K-1})$.

(Case 4) $\rho \geq R(K, a_2)$: $\theta_1 = -1$.

Here $b_1 = (K-1)(1 + e^{-2} - 2e^2)a_2 - 2$ and $b_2 = 8(1 + e^{-2})(K-1)a_2((K-1)a_2 - e^2)$. $R(K, a_2)$ is given by:

$$R(K, a_2) = \frac{-b_1 + \sqrt{b_1^2 + b_2}}{2(1 + e^{-2})a_2}. \tag{71}$$

$\square$

### B.3 PROOF OF REMARK 2

*Proof.* Recall that:

$$R(K, a_2) = \frac{-b_1 + \sqrt{b_1^2 + b_2}}{2(1 + e^{-2})a_2}. \tag{72}$$

where $b_1 = (K-1)(1 + e^{-2} - 2e^2)a_2 - 2$, $b_2 = 8(1 + e^{-2})(K-1)a_2((K-1)a_2 - e^2)$. $b_1$ and $b_2$ in Eq. (72) can be roughly simplified as

$$
\begin{aligned}
\frac{b_1}{a_2} &= (K-1)(1 + e^{-2} - 2e^2) - \frac{2}{a_2} \approx (K-1)(1 + e^{-2} - 2e^2) = (K-1)b_1' \\
\frac{b_2}{a_2^2} &= 8(1 + e^{-2})(K-1)((K-1) - \frac{e^2}{a_2}) \approx 8(1 + e^{-2})(K-1)^2 = (K-1)^2 b_2',
\end{aligned} \tag{73}
$$

where $b_1' = (1 + e^{-2} - 2e^2)$ and $b_2' = 8(1 + e^{-2})$. Then we can roughly simplifies $R(K, a_2)$ as a function only respect to $K$ as:

$$
\begin{aligned}
R(K, a_2) &= \frac{-b_1 + \sqrt{b_1^2 + b_2}}{2(1 + e^{-2})a_2} = \frac{-\frac{b_1}{a_2} + \sqrt{(\frac{b_1}{a_2})^2 + \frac{b_2}{a_2^2}}}{2(1 + e^{-2})} \\
&\approx (K-1)\frac{-b_1' + \sqrt{{b_1'}^2 + b_2'}}{2(1 + e^{-2})} \\
&= (K-1)\frac{-(1 + e^{-2} - 2e^2) + \sqrt{(1 + e^{-2} - 2e^2)^2 + 8(1 + e^{-2})}}{2(1 + e^{-2})} \\
&= R'(K) \\
&\approx 12.16(K-1)
\end{aligned} \tag{74}
$$

$\square$

