# OpenReview forum: "Geometry of Long-Tailed Representation Learning: Rebalancing Features for Skewed Distributions"
_ICLR.cc/2025/Conference — ICLR 2025 Poster_

### Official Review · Reviewer_nqrn · 2024-10-21

**Soundness:** 3
**Presentation:** 3
**Contribution:** 2
**Rating:** 6
**Confidence:** 5

**Summary:**

This paper aims to address the long-tailed recognition problem.
The paper provides a sound theoretical analysis of contrastive learning from the perspective of neural collapse.
Based on the analysis, they draw the conclusion that increasing the size of samples for tail classes can alleviate the collapse issue.
Thus, the proposed method is to synthesize sample features for tail classes.
Moreover, a class-wise temperature schedule is used in their algorithm.

Experiments are conducted on popular long-tailed benchmarks, like ImageNet-LT, iNaturalist, and CIFAR-LT. Some improvements are observed when compared with baselines, like KCL, TSC, and BCL.

**Strengths:**

(1) Sound theoretical analysis on neural collapse for contrastive learning.
(2) The paper is well-organized and easy to follow.

**Weaknesses:**

(1) Lack of comparisons with current state-of-the-art methods, like GPaCo, especially on large-scale data including ImageNet-LT and iNaturalist 2018.

[ref1] Generalized Parametric Contrastive Learning, TPAMI 2023.

(2) Synthesizing sample features for tail classes is not a novel idea. The paper should discuss the differences between the proposed method and previous relevant works, like ReMix and work[ref2].

[ref2] Feature Space Augmentation for Long-Tailed Data, ECCV 2020.

(3) The connection between the proposed method, like feature augmentation and temperature adjustment, and the neural collapse theory is not strong. Increasing the samples for tail classes can obviously alleviate the long-tailed recognition problem.

(4) The claim that "Head classes benefit from a larger τ while tail classes benefit from a smaller one. We demonstrate
(in Appendix A.3.2) it holds for SC loss too." seems to contradict Eq. (11) that the tail classes will be assigned with a large temperature.

(5) Minor typo, "N_min - N_min" should be  "N_max - N_min" in Eq. (11).

**Questions:**

(1) Fair comparisons with current state-of-the-art methods, like GPaCo [ref1], on large-scale datasets including ImageNet-LT and iNautralist 2018.

(2) Discussion on the differences between the proposed method and previous augmentation-based methods, like ReMix and work[ref2].
     What are the advantages of the proposed method over them?

(3) Clarification on the temperature adjustment.

---

### Official Review · Reviewer_MXe4 · 2024-10-31

**Soundness:** 3
**Presentation:** 4
**Contribution:** 3
**Rating:** 8
**Confidence:** 3

**Summary:**

* _Theoretical contributions_: The paper introduces a theoretical framework to study how imbalanced/long-tailed (LT) data distributions distort class representations in feature space. It mathematically demonstrates that feature centers of underrepresented classes (tail classes) collapse or shrink, leading to overlapping regions that make class separation challenging.

* _Technical contributions_: The paper proposes a new method, FeatRecon, which reconstructs the feature space to enforce symmetry and linear separability among classes. FeatRecon achieves this by generating _synthetic features_ for underrepresented classes and iteratively adjusting class representations to counteract the collapse effect. It introduces a novel approach to estimate and regularize "confidence supports", ie regions in feature space for tail classes using head class statistics.

* _Experimental evaluation_: FeatRecon demonstrates state-of-the-art performance on the common LT benchmarks, with strong accuracy improvements in places, eg for the more challenging iNat dataset.

**Strengths:**

The paper offers an analysis on imbalanced data feature space geometry for the case of the SCL loss. It presents a novel and technically sound theoretical framework that motivates a novel method. The analysis in Sec 2 is complemented with clearly drawn figures/ilustrations. The paper is clearly written and has strong experimental validation supporting the new method.

**Weaknesses:**

W1: In the proposed method,  the regularize tail classes’ estimation of statistics using the statistics of neighboring head classes. The intuition behind that choice is not clear to me.

W2: There seems to be a typo in Table 5: loss in row 3 is relative to row 2, but gains in rows 4 and 5 are relative to row 1.

**Questions:**

Q1: In Sec 3.2 "optimal" representation is defined wrt to training loss. It is unclear to me how this correlates to performance for the case of LT, where at test time we care equally about accuracy over all classes, ie what is defined as  "distribution shift." in 4.1. It would be nice to discuss in the theoretical part how optimality at training in this case is not the same as optimality for recognition accuracy, where the average is over classes.

Q2: You now sample uniformly in the estimated confidence support region. Would it make sense to sample that support space better? e.g. methods like MoCHi [Hard negative Mixing, NeurIPS 2020] generate synthetic features for contrastive learning by interpolating neighbouring features in feature space. Interpolating tail class features using the neighbouring head class features would result in a similar generation (eg in Fig 4). Could be an interesting sampling to ablate.

Q3: How sensitive is regularisation to the hyper parameters like q (ie top-q head classes selected)?

---

### Official Review · Reviewer_2QQJ · 2024-11-02

**Soundness:** 3
**Presentation:** 3
**Contribution:** 3
**Rating:** 6
**Confidence:** 4

**Summary:**

This work proposes a method based on contrastive learning to address the long-tailed learning problem. By uniformly sampling the confidence supports of the corresponding classes, a consistent number of samples for each class is achieved for contrastive learning. Additionally, the method incorporates the LDAM loss to facilitate supervised learning with long-tail data.

**Strengths:**

1. The authors provide a comprehensive theoretical framework.
2. offers an intriguing method for sampling features with a balanced number of categories and provides a solid theoretical foundation.

**Weaknesses:**

1. Feature Visualization Lacks Clarity: The feature visualization presented in the manuscript is insufficient. For instance, Figure 4 appears to be a diagram created by the authors. I would like to see whether the actual training results align with expectations. Could the authors provide additional visualization results, such as t-SNE, to illustrate the training process more clearly?

2. Distribution Changes Despite Fixed Empirical Centers: In Figure 4, if the Empirical Centers remain unchanged, what accounts for the changes in distribution? In layman's terms, the class to which a sample belongs is ultimately determined by the "distance" of the sample from the class center.  Or maybe the expression regarding Real Distributions is somewhat unclear. I do not fully understand why the real distribution changes and how it should be aligned with confidence supports.  Specifically, what does this refer to? Is it the ground-truth distribution? If the results are as shown in Fig.4 (a), the training samples appear to be completely separated. If only the existing training data is used, the situation seems no different from Fig.4 (d), and the final classifier would not change. Could the authors provide further clarification on this matter?

3. There is a lack of related work, and some important and latest work is missing.
For example,

Decoupling methods:

[1] B. Zhou, et al., BBN: Bilateral-Branch Network with Cumulative Learning for Long-Tailed Visual Recognition, in CVPR 2020.

[2] Z. Zhong, et al. Improving Calibration for Long-Tailed Recognition, in CVPR 2021.

[3] M. Li., et al., Feature Fusion from Head to Tail for Long-Tailed Visual Recognition, in AAAI 2024.

Logit adjustment methods:

[4] J. Ren. et al., Balanced Meta-Softmax for Long-Tailed Visual Recognition, in CVPR 2020.

[5] M. Li., et al., Long-tailed visual recognition via gaussian clouded logit adjustment, in CVPR 2022.

Transfer learning (learving pre-trained transformer-based backbone):

[6] J. Shi et al. Long-Tail Learning with Foundation Model: Heavy Fine-Tuning Hurts, in ICML 2024.

It is recommended that the author conduct a comprehensive analysis of related work. Or conduct a more detailed analysis of the long-tail learning method based on contrastive learning. However, important work should not be lacking.

4. Tiny issue:
A. K. Menon, et al., Long-tail learning via logit adjustment. ICLR 2021. The work is publushed in ICLR 2021, not 2020.

**Questions:**

See Weaknesses 1 and 2.

---

### Official Review · Reviewer_itNQ · 2024-11-03

**Soundness:** 3
**Presentation:** 3
**Contribution:** 3
**Rating:** 6
**Confidence:** 3

**Summary:**

This paper presents a theoretical analysis and solution for the performance degradation of deep learning models when faced with long-tailed data distributions, introducing a method called FeatRecon. The authors provide a systematic exploration of how long-tailed data skews feature representations and detail the resulting limitations on the model's generalization capabilities. Their findings indicate that feature centers of tail classes may shrink and overlap, leading to inseparable representations across different classes.

To address this issue, the authors propose an algorithm that generates synthetic features to balance the sample sizes of all classes, constraining these synthetic features within confidence supports estimated from head class statistics. Furthermore, their iterative approach aims to create a symmetric and linearly separable feature space, thereby enhancing the model's robustness to long-tailed data. Extensive experiments conducted on multiple long-tailed datasets, including CIFAR-10-LT, CIFAR-100-LT, ImageNet-LT, and iNaturalist 2018, demonstrate the effectiveness of the proposed method.

Overall, this work effectively bridges theoretical insights and practical applications, offering valuable contributions and innovative techniques for tackling long-tailed data challenges, thereby possessing significant research merit.

**Strengths:**

This paper has remarkable advantages. In terms of innovation, the theoretical contributions are prominent, including in-depth analysis of the impact of long-tailed data on feature representation and the construction of a theoretical framework for optimal representation configuration; at the same time, the methodological innovation is unique, and the FeatRecon method effectively solves related problems by reconstructing the feature space. The experimental design is elaborate, with extensive and comprehensive datasets, detailed and sufficient comparisons, and reasonable ablation studies. The result analysis shows significant performance advantages and in-depth and detailed discussions.

**Weaknesses:**

1. Theoretical Basis

Limitations of Theoretical Assumptions: The theoretical analysis mainly relies on a simple one - vs - all scenario assumption, which involves adjusting the sample size of one category while keeping the sample sizes of the others equal and fixed. This assumption may have certain limitations in practical applications and fails to fully cover the complex distribution of long - tailed data.

The Inadequate Combination of Theory and Practice: Although a theoretical framework is proposed in the paper, in practical applications, the connection between some designs of the FeatRecon method and the theoretical analysis is not direct and close enough. For example, in the processes of synthetic feature generation and confidence support estimation, further elaboration is needed on how to better reflect the analysis of category centers and feature distributions in the theory.

2. Experimental Comparison

Incomplete Comparison with the Latest Methods: Although the paper has compared with numerous existing methods, in some cutting-edge research directions of long-tailed learning, there might be some of the latest methods not included in the comparison experiments, making it impossible to fully determine the absolute superiority of the proposed method in the entire research field.

The Issue of Consistency in Experimental Settings: There are some differences in the experimental settings for different datasets, such as different backbone networks, varying numbers of training epochs, and diverse data augmentation methods. Although the author may aim to adapt to the characteristics of different datasets, this inconsistency could have a certain impact on the comparison and analysis of the results, and the rationality of different settings requires further explanation.

3. Writing Standards

Complexity in Some Content Representation: In certain sections of theoretical derivations and method descriptions, the paper presents a relatively high level of complexity. It employs a substantial number of mathematical symbols and formulas, thereby augmenting the difficulty for readers to comprehend. For instance, during the proof of Theorem 2, some derivation steps could be elaborated more thoroughly to assist readers in better following the author's line of reasoning.

**Questions:**

1. Theoretical Aspects

Expand the Theoretical Framework: Consider relaxing the one - vs - all assumption and researching more complex long - tailed data distribution situations to further enhance the theoretical framework and make it more versatile and practical.

Strengthen the Connection between Theory and Method: In the method description section, more explicitly expound on how to design and optimize the method according to theoretical analysis. For instance, in the processes of synthetic feature generation and confidence support estimation, clarify how to utilize theoretical results to guide the selection and adjustment of parameters, thus better integrating theory and method.

2. Experimental Aspects

Supplementary Comparative Experiments: Pay attention to the latest research advancements in the field of  long - tailed learning and add comparative experiments with more recent methods to evaluate the performance advantages and innovativeness of the FeatRecon method more comprehensively.
Unify Experimental Settings: Whenever possible, strive to unify the experimental settings across different datasets as much as possible, or provide a detailed explanation for the reasons behind different settings, in order to compare and analyze the experimental results more clearly.

3. Writing Aspects

Simplify the Expression: Simplify and elucidate the complex theoretical derivations and method descriptions. Employ more textual explanations to aid in understanding mathematical formulas and symbols, thus avoiding readers from getting lost while reading.

---

> ### Comment · Reviewer_itNQ · 2024-11-26
> **Thanks for the feedback from the authors**
>
> Thank you for the additional discussion on the theoretical analysis and the comparisons with PaCo and GPaCo. However, I maintain my original score, as the improvements over GPaCo appear to be marginal.

---

### Meta-Review · Area_Chair_jWbL · 2024-12-22

**Metareview:**

The reviewers were unanimously positive about this paper, and its strengths in theoretical innovation, methodological rigor, and experimental validation outweigh the identified weaknesses. While the concerns raised highlight areas for improvement, they do not detract significantly from the overall quality and contributions of the work. The AC panel decided to accept the paper.

The authors are encouraged to address the reviewers' comments in the final camera-ready version to strengthen their presentation further. Specifically, including additional experimental comparisons, clarifying theoretical-practical connections, and enhancing visualization results would add significant value.

**Additional Comments On Reviewer Discussion:**

Reviewer itNQ retained the borderline accept score stating the marginal improvement compared to GPaCo. Reviewer nqrn also felt this paper offered only marginal improvement against GPaCo, but was leaning towards acceptance for the theoretical contribution of neural collapse in contrastive learning.  Reviewers 2QQJ, MXe4   were happy that it addressed their primary concerns and was quite positive about the paper.

---

### Decision · Program_Chairs · 2025-01-22

Accept (Poster)